# The geology and evolution of the Near-Earth binary asteroid system (65803) Didymos

Images collected during NASA's Double Asteroid Redirection Test (DART) mission provide the first resolved views of the Didymos binary asteroid system. These images reveal that the primary asteroid, Didymos, is flattened and has plausible undulations along its equatorial perimeter. At high elevations, its surface is rough and contains large boulders and craters; at low elevations its surface is smooth and possesses fewer large boulders and craters. Didymos' moon, Dimorphos, possesses an intimate mixture of boulders, several asteroid-wide lineaments, and a handful of craters. The surfaces of both asteroids include boulders that are large relative to their host body, suggesting that both asteroids are rubble piles. Based on these observations, our models indicate that Didymos has a surface cohesion $\leq 1$ Pa and an interior cohesion of $\sim 10$ Pa, while Dimorphos has a surface cohesion of $<0.9$ Pa. Crater size-frequency analyzes indicate the surface age of Didymos is 40–130 times older than Dimorphos, with likely absolute ages of $\sim 12.5$ Myr and $<0.3$ Myr, respectively. Solar radiation could have increased Didymos' spin rate leading to internal deformation and surface mass shedding, which likely created Dimorphos.

The NASA Double Asteroid Redirection Test (DART) spacecraft successfully struck Dimorphos, the moon of the binary asteroid (65803) Didymos, on Sept. 26, 2022[1]. Prior to colliding with the surface, the Didymos Reconnaissance and Asteroid Camera for OpNav (DRACO)[2] imaged both Didymos and Dimorphos. These data were followed by images of Didymos and Dimorphos collected by the Italian Space Agency's (ASI) LICIACube Unit Key Explorer (LUKE) several 10 s to 100 s after impact[3]. Although the encounter possessed some viewing limitations, these images provide the first well-resolved look of a small near-Earth binary asteroid system.

Telescopic observations of the near-Earth asteroid (NEA) population have shown that a sizeable fraction exist as binary systems[4]. These asteroid binaries are hypothesized to form when increases to the spin rate of a single rubble-pile asteroid causes it to mechanical fail, details of which depends on the asteroid's geophysical properties[5]. The shape and surface morphology of asteroids are key to interpreting their evolution. However, while previous spacecraft observations provided insights into the geological characteristics of single asteroids such as Bennu and Ryugu e.g., [6,7], the proximity observations of the

Didymos system collected by the DART mission provide a unique opportunity for a close-up geological look of an NEA binary system from which we can infer its geophysical properties and expand our understanding on their formation.

Here, we analyze the obtained images and digital terrain model developed therewith to provide constraints on the physical nature of these asteroids and on the geological origin and evolution of the Didymos system. In brief, our observations show that Didymos is flattened relative to other similar-sized asteroids and may have waviness along its equatorial perimeter. Its polar regions are rough and covered with large boulders, craters, and/or a plausible trough, while near its equator, Didymos' surface is smooth, with few large boulders and craters. Dimorphos possesses an intimate mixture of boulders of all sizes, several asteroid-wide cracks or faults, and a handful of craters. The surfaces of both asteroids include boulders that are large relative to their host body, suggesting rubble-pile interior structures. Spin-up models indicate that to match observations, Didymos' surface material is weak at $\leq 1$ Pa, but its interior is slightly stronger at $\sim 10$ Pa. Likewise, Dimorphos probably possesses weak surface material at $<0.9$ Pa. Using

✉ e-mail: olivier.barnouin@jhuapl.edu

these surface strengths and the number of observed craters, we find that Didymos' surface is 40–130x older than Dimorphos, with likely absolute surface ages of ~12.5 Myr and <0.3 Myr, respectively. The evidence is consistent with radiatively-driven spin-up causing internal deformation and limited surface mass shedding of Didymos to form Dimorphos.

## Results

### The shapes of Didymos and Dimorphos
The shapes of Didymos and Dimorphos are key to interpreting their evolution. Didymos's shape is smaller along the polar axis relative to an earlier radar-derived shape model[8] and other "spinning top-shaped" asteroids (e.g., Bennu, Ryugu) that have been previously visited by spacecraft (Fig. 1). Using best-fit ellipsoids, Didymos has an intermediate to major axis ratio $b/a = 0.96$, and a minor to major axis ratio $c/a = 0.73$. A cluster of generally top-shaped asteroids, which includes Bennu and Ryugu, sits near $b/a \approx 0.95$, and $c/a \approx 0.9$. We note, however, that radar shapes have larger uncertainties, usually along their z-axes, depending on the viewing geometry. Didymos possesses a ridge at, or near, its equator that is nearly triangular in shape (Fig. 2a). Unlike Ryugu or KW4[9], Didymos' perimeter when viewed somewhat obliquely (Fig. 2b) or along its south pole (Fig. 2f) is not particularly circular, a trait that is reminiscent of Bennu, which has longitudinal (N-S) ridges that are apparent when Bennu is viewed from the same direction[6]. The squashed appearance of Didymos is broken by a few large protruding boulders that are observed on its limb, usually at moderate to high (>15°) latitudes. Some of the boulders are quite large relative to the overall size of Didymos, with the largest measuring ~160 m across (Fig. 2b) or ~21% of Didymos' diameter.

Dimorphos' uniform oblate ($b \approx a$) spheroid[1] shape may be unusual relative to other secondaries (Fig. 1), which are more typically elongated with $0.7 < b/a < 0.9$, though there may be observational biases toward elongated objects[10]. Like for Didymos, Dimorphos' profile is broken up by larger boulders (>10 m in effective diameter) strewn across the asteroid's surface (Fig. 3).

### The surface geology of Didymos
Didymos can be divided into three geological units (Fig. 2c). In the limited views available of Didymos, regions at higher latitudes (poleward of 20°) are undulating and rough, though these rough

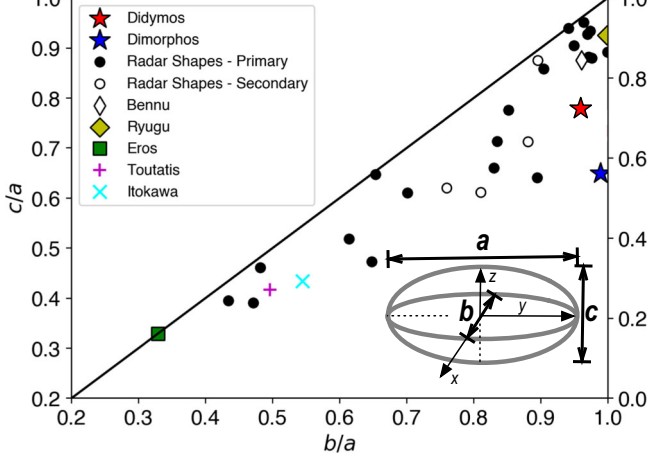

**Fig. 1 | Aspect ratios of asteroids visited by spacecraft and observed by radar.** The intermediate to major axis ratio, $b/a$, and minor to major axis ratio, $c/a$, shows that Didymos possesses a smaller $c/a$ relative to other top-shaped asteroids. Dimorphos has a large $b/a$ relative to other secondaries. The four secondaries included are the satellites of the S-type binaries 66391 Moshup (formerly 1999 KW₄) and 2000 DP₁₀₇ as well as the two satellites of the C-type triple 2001 SN₂₆₃ [83–85]. The black line defines $b/a = c/a$.

regions extend towards the equator east of the prime meridian. These regions possess many large boulders (10–160 m in length)[11], degraded craters (up to 270 m in diameter) and/or broad troughs. At lower latitudes between ~10°S and ~10°N, and west of ~20°E, and, the terrain appears smooth at the resolution of the available images and lacks craters. This "smooth" region could be dominated by rocks and boulders that are smaller than the 5-m pixel scale of the best images and resembles what is seen in DRACO images of Dimorphos with comparable pixel scales. Preliminary albedo assessments[12] indicate that this smooth terrain is darker than the surrounding terrain. Transition regions separate Didymos' rough and undulating terrain from the smooth terrain on either side of the equator, showing evidence for regolith mass movements (Fig. 2a), including possible boulder tracks[13]. The terrains identified are consistent with the spatial distribution of surface roughness measured on Didymos[14].

Didymos exhibits an exceptionally bright (~20% brighter than average[15];) spot within the transition region close to the subsolar point (Fig. 2a). This spot is close to the rim of one of the candidate craters and otherwise is not associated with any other prominent topographical or geological anomaly.

To some extent, Didymos' geology reflects the effects of surface elevation[7] (Methods [methods:surf]). The rough terrain corresponds to highlands, while smooth terrains are lowlands, very reminiscent of what has been observed at Itokawa[16]. The importance of elevation was also noted at the fast-rotating Bennu (~4.3 hr)[6], but is less evident at slow-rotating Ryugu (~7.63 hr)[17].

### The surface geology of Dimorphos
Dimorphos has no distinct geological units and its overall albedo is slightly brighter than the average albedo of Didymos[12]. Its surface is covered by an intimate mixture of boulders[11] (Udden-Wentworth scale; > 1 m-in-length) and cobbles (6–100 cm-in-length; Figs. 3 and 4). The largest of the observed boulders is 16 m in length, too big to have been formed by the impacts that formed craters observed on Dimorphos, which are either similar or smaller sized than this length. No evidence exists for large collections of fine particles with sizes below the pixel scale of the finest-resolution complete image (5.5 cm). Regions that might be construed as fine-grained in lower-resolution (> 100 cm/px) images are revealed in higher-resolution images (< 10 cm/px) to be piles of smaller rocks. The counts of boulders on Dimorphos >2 m-in-length are considered statistically to be complete[11], and the total number of boulders >2 m in size per 10 degree bin shows no strong dependencies with latitude or longitude (Fig. 5).

Many of Dimorphos' boulders possess evidence for linear cracks on their surfaces (Fig. 4a). Some of these cracks are likely the product of thermal fragmentation[18]. Multiple parallel lineaments are visible on a few individual rocks (e.g., Fig. 4c). The presence of these lineaments could be evidence for either layering, shatter cone-like structures, or albedo markings that could be due to shock[12]. While shatter cones are typically found in fine quartzite and dolomite, they have been identified in basalt[19], which is similar to the likely ordinary-chondrite composition of Dimorphos. Some of the boulders possess plausible craters (Fig. 4b), and there may be a camp-fire structure (Fig. 4d) interpreted to be the result of boulder disruption from impact[20] or thermal fragmentation.

Some individual rocks and boulders sit atop other boulders (Fig. 4c); however, many rocks lay alongside larger ones (Fig. 4c). This latter observation could be evidence for imbrication[21] resulting from surface boulders migrating across the surface of Dimorphos systematically as a result of slope processes. However, these boulders are not preferentially located along one side of the other boulders, nor are they orientated with their long axes in the slope direction (Fig. 6) as would be expected if they had slid downslope. Our preferred interpretation, therefore, is that many of these boulders leaning or laying on other boulders are local debris aprons formed when rocks slid off

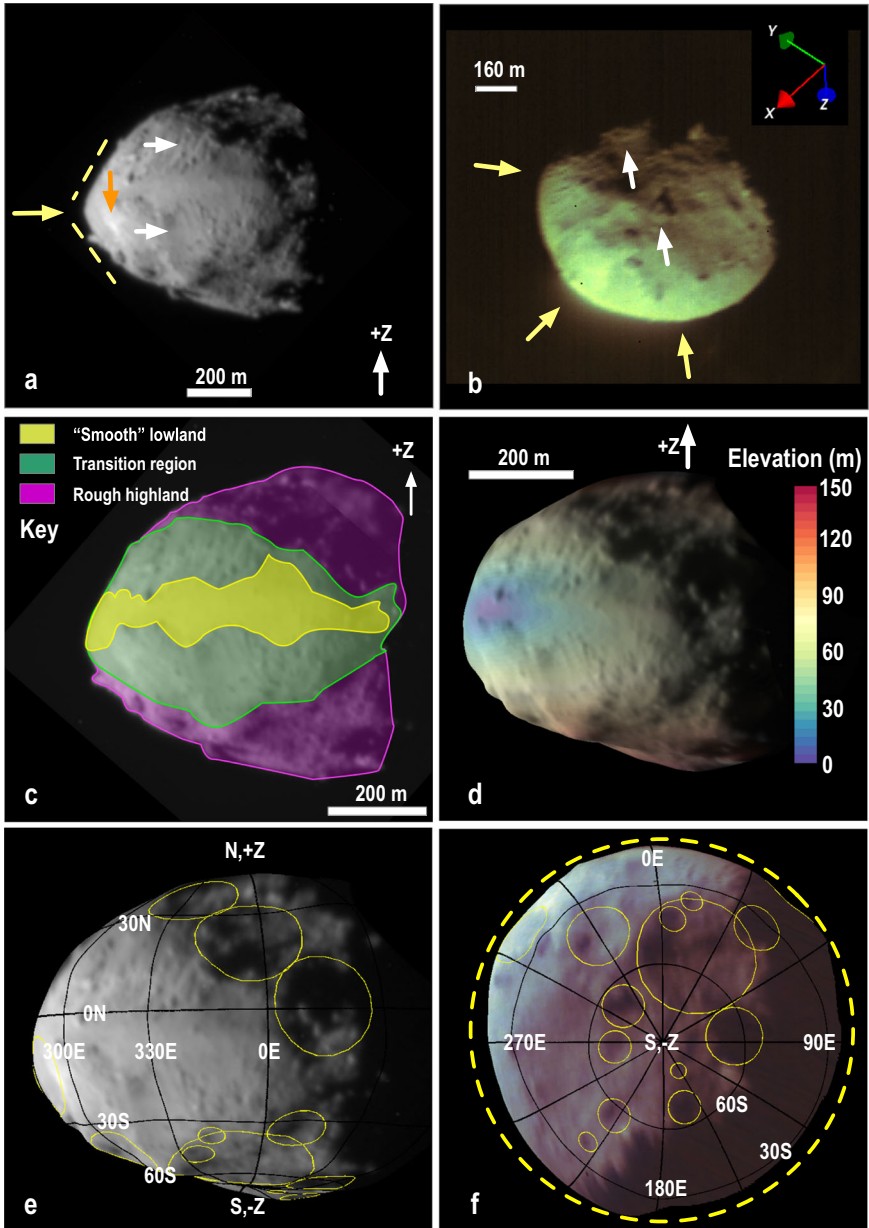

**Fig. 2 | Geological properties, craters and shape of Didymos.** Images (**a**–**c**), geological units (**c**), shape (**d**, **e**), elevation (**d**), and crater candidates (**e**, **f**) of Didymos. **a** DRACO image (dart_0401929893_44497) highlights the triangular shaped ridge (yellow arrow and dashed lines), protruding boulders, boulders with north-south tracks (upper white arrow), and an area with evidence for a plausible mass movement (lower white arrow). The orange arrow indicates the location of the bright spot. **b** LUKE image (liciacube_luke_l2_1664234227_01003) indicates plausible corners or longitudinal N-S ridges. The solid white arrows point to 160-m and 85-m boulders. Units shown on mosaic (**c**; dart_0401929893_44497 and dart_0401929913_07357) may be the result of elevation changes, overlain here on the Didymos GDTM (**d**). Crater candidates are shown on the GDTM in (**e**) and (**f**). Consistent with (**b**), GDTM (**f**) shows some non-circular perimeter attributes when viewed from -Z pole. GDTM does not include limb data, and thus lacks the triangular outline seen in the images (**a**–**c**; see Methods).

larger boulders and accumulated by their sides with no preferred orientation. These rock accumulations do not indicate widespread regional regolith transport.

The debris-apron interpretation is consistent with shallow (<10°) average surface slopes of Dimorphos (Fig. 6). When the roughness generated by the boulders is accounted for in the imaged sections of Dimorphos (Fig. 6b), the slope vectors point in random directions. It is unsurprising that smaller rocks on Dimorphos accumulate in debris aprons around bigger boulders rather than generate imbrication features. The low average global slope and the dominance of local roughness in minimizing local regional surface displacements further suggest that Dimorphos' current shape and surface are stable, and

likely have undergone only minimal changes between the time they achieved this configuration and the DART impact.

Dimorphos shows a few long, noticeable lineaments. A broad trough spans the northern portion of the asteroid. Long and narrow fractures are visible and are usually linear, although one of the most obvious ones is curved. In all cases, these narrow lineaments span significant fractions of the southeastern part of Dimorphos that was observed with DRACO (Fig. 4e, f). These lineaments are not obviously related to any crater candidate seen on the surface. In the best images collected, there also exists a linear fabric (Fig. 4d). Some of the smaller boulders align with it (Fig. 4c) but not in the slope direction[11].

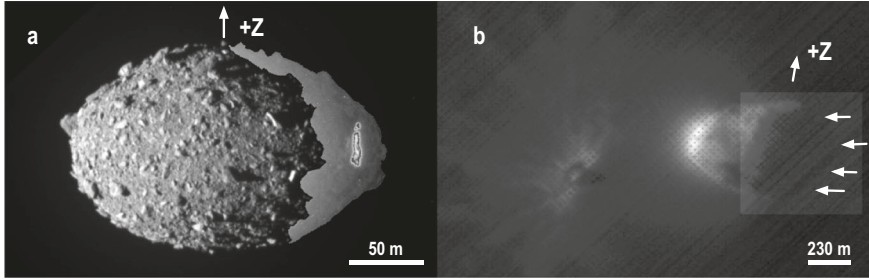

**Fig. 3 | Illuminated and backlit outlines of Dimorphos and Didymos.** Composite images of Dimorphos (DRACO image collected on Sept 26, at 23:14:09 UTC; **a**) and Didymos (LUKE images collected on Sept 26, at 23:17:27; **b**). The images are composites of two stretches to show the full outline of the bodies. Regions of Dimorphos illuminated by Didymos shine appear light gray in (**a**). Arrows in (**b**) point to the dark limb of Didymos back-lit by DART ejecta. In (**b**), the +Z (N) direction is tilted by a few 10 s of degrees into the page. Dimorphos and DART ejecta can be seen in (**b**) at roughly 8:30 clock position from the center of Didymos.

## Craters on Didymos and Dimorphos

We identify sixteen plausible craters on Didymos. They were identified from DRACO and LUKE images, guidance from a global digital terrain model (GDTM) (Methods [methods:didymos]), a set of reasonable observational criteria (Methods [methods:didymos_crater]), and consensus from several authors. Most crater candidates are located at higher latitudes. The largest ones (>160 m-in diameter) are degraded and resemble broad circular depressions seen on asteroids Ida (S-type)[22], Eros (S-type)[23], Itokawa (S-type)[24], and Bennu (C-type)[25]. The general morphology of these larger circular depressions is probably due to the influence of the asteroid's shape on crater growth[25]. The smaller craters possess a more classical bowl shape. Evidence for well-preserved, well-defined crater rims and ejecta is difficult to discern in the available images.

The DRACO images and a GDTM of Dimorphos permitted identifying twelve plausible craters on this asteroid using a methodology that differed from what was done at Didymos. Initial identification was performed by inspecting DRACO images projected onto the Dimorphos shape model using the Small Body Mapping Tool (SBMT)[26], yielding >22 candidate craters. These candidate craters were then vetted by evaluating their depths, diameters, and overall shapes using local DTMs (Methods [methods:dimorphos_crater]).

The identified craters are similar to those observed on other asteroids with boulder-dominated surfaces[24,27,28]. They are depressions filled with a range of smaller rocks—which may represent disrupted material—surrounded by an uneven rim dominated by individual boulders. The mean geometric[29] depth-to-diameter ratio of $0.152 \pm 0.04$ for craters on Dimorphos is similar to that found on the C-type Bennu ($0.17 \pm 0.04$; see Supplementary Fig. 2), but deeper than on C-type Ryugu[27] for craters of similar size Fig. 7). We caution against making any conclusive interpretations regarding these similarities or differences, as the quality of the underlying datasets used to produce the crater DTMs here are not as good as those for Bennu and Ryugu. One of the craters identified presented evidence of a central mound (crater crt10 in Supplementary Table 4). The material comprising the mound appears to be an accumulation of bright and dark boulders. The other craters show no obvious evidence for either a bench or mound.

The crater cumulative size frequency distributions (CSFD) on both Didymos and Dimorphos (Fig. 7) indicate that Didymos has a higher density of craters and thus an older surface. We provide a range of plausible surface ages for both asteroids, using cratering strengths from $Y = 10 - 10^5$ Pa based on experimental data that have been used to construct crater-scaling relationships[30] and supplemented with recent insight into the material strength of Dimorphos based on analysis of the outcome of the DART impact[31,32]. The ages were calculated using the general form of the crater scaling relationship that includes both strength and gravity contributions with crater scaling constants for sand and cohesive soils[30]. We assume a main asteroid belt (MBA) impactor population for Didymos and a mixed population of NEA and MBA impactors for Dimorphos (see Methods [methods:age]).

## Inferred strength properties of Didymos and Dimorphos

We combine geophysical evidence from our surface observations with insights from the outcome of the DART impact experiment[31,32] to better constrain the inferred ages of both Didymos and Dimorphos to understand the origin and evolution of these bodies. While there are many definitions of surface strength as applied to rocks, including compressive, tensile strength, and bulk cohesion $C$, we focus on $C$ and take it to be equivalent to cratering strength, $Y$, because they are likely similar in magnitude[32]. $Y$ is a key input needed to define surface ages.

Didymos's surface cohesion, $C$ can be estimated using a Factor of Safety (FS) analysis that was employed at Bennu[33]. This FS analysis provided a good match to surface cohesion estimates derived from analyzing interactions between the OSIRIS-REx Touch-And-Go sampling device and Bennu's surface[34,35]. Assuming our current estimate of the density of Didymos ($2800 \pm 280$ kg/m³), the rapid 2.26 hr rotation period of Didymos leads to steep slopes between $\sim 10°$ and $\sim 30°$ latitude (Fig. 8). Assuming that the surface material has friction angles of 35° as inferred on Dimorphos (see discussion), the FS analysis indicates that these slopes should see landsliding of material $\leq 10$ m in thickness with near-surface $C \leq 1$ Pa. We choose a 10 m-thick layer because this is approximately the thickness of the more easily mobilized layer on Bennu and Ryugu[33,36,37], and when spread evenly across Didymos, provides enough material to generate $\sim 10$x the volume of Dimorphos, far more than needed based on current estimates needed for the moon's formation[38]. Thinner layers and a lower bulk asteroid density would require less cohesion[33], while thicker layers and larger densities (within $\pm 280$ kg/m3) would require more cohesion but always < 2 Pa. For our nominal assumptions (density of 2800 kg/m³ and 10 m thick regolith layer), surface $C > 1$ Pa would shut down surface displacements completely. The starts of a few boulder tracks and the source region of one of plausible landslide coincide well with regions where FS < 1 when $C \sim 0.5$ Pa (Fig. 8). The maximum allowable $C$ of 1–2 Pa is similar to cratering strength values estimated at the asteroid Ryugu[36,39] and Bennu[35]).

While a near-perfect circular equatorial perimeter would indicate significant interior deformation of a fairly malleable object[7], the plausible presence of undulations in Didymos' equatorial perimeter, which may be N-S ridges, would indicate that Didymos contains large distinct mass concentrations (large blocks) that influence the asteroid's shape, perhaps analogous to findings at Bennu[40]. These mass concentrations likely require some strength to permit the perimeter to protrude in some locations. Other possible explanations for such protrusions and longitudinal (N-S) ridges at Bennu include structural wedges that resulted from an early quasi-disruption of the asteroid[41] from spin-up.

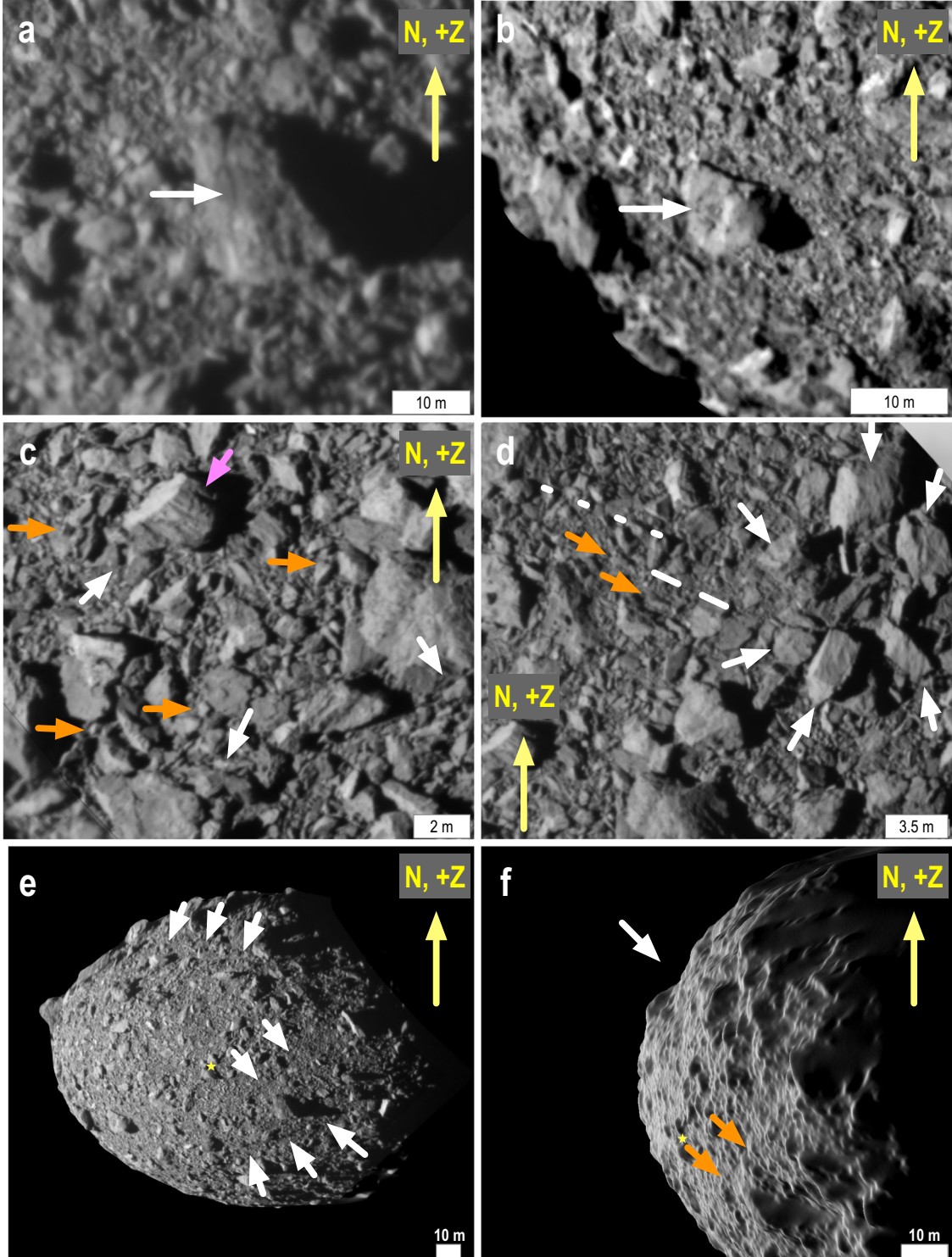

**Fig. 4 | Geological features on Dimorphos.** Arrows indicate a boulder with fractures (**a**), a crater on boulders with evident spall (**b**), randomly orientated debris aprons (white arrows; **c**) rocks on boulders (orange arrows; **c**) multiple lineaments on a single boulder (magenta arrow; **c**), a possible camp-fire structure (white arrows; **d**) and a linear fabric (dashed lines; **d**) with aligned boulders (orange arrows; **d**) and surface lineaments (**e**, **f**). All images were collected by DRACO moments before the DART spacecraft collided with Dimorphos. The global view (**e**) shows the location of the DART impact (star), an asteroid-wide trough (upper set of arrows) and several asteroid-wide fractures. The trough (white arrow) and one of the fractures (orange arrows) are evident in oblique-views (**f**) of Dimorphos' global digital terrain model[10].

Such a scenario, however, was found to be unfeasible[42]: the forces needed to lead to quasi-failure were so large that the asteroid would have disrupted completely. This would be likely true for Didymos as well.

The presence of 10 Pa strength within Didymos is consistent with the evidence for surface mass-movements and mass shedding. Several studies show that asteroid spin up facilitates mass movement and mass shedding, especially for lower porosity objects like Didymos, if the

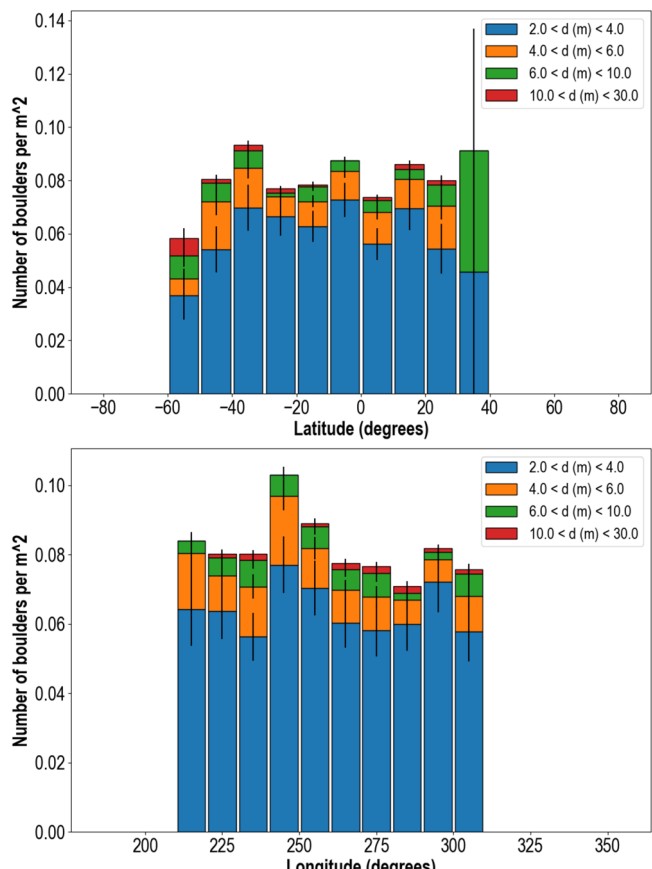

**Fig. 5 | Spatial distribution of boulders on Dimorphos.** Total boulder distribution as a function of latitude (**a**) and longitude (**b**) on Dimorphos, for different boulder sizes. Errors bars show 1-σ uncertainty.

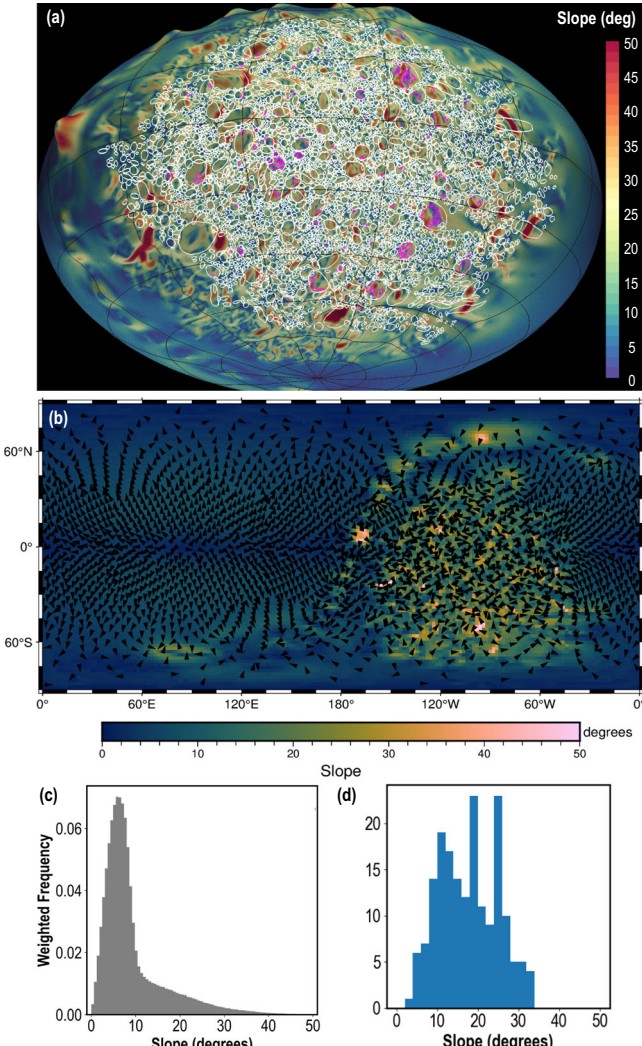

**Fig. 6 | Global slopes relative to gravity on Dimorphos.** All slopes include the gravitational effects of Didymos and Dimorphos's assumed synchronous rotation. **a** Locations of boulders on Dimorphos (ellipses). The magenta ellipses indicate boulders located on other boulders. **b** A global slope map of Dimorphos with arrows showing the downslope direction. Regions that show no roughness were not part of the sunlit terrain seen by DART. **c** The global distribution of slopes weighted by surface area. The slope tail (>10° in (**c**)) is dominated by the surface boulders; the rest of the distribution reflects the overall oblate ellipsoidal shape of the asteroid. **d** The distribution of boulders on other boulders, which exist on slopes <35°.

interior possesses some strength[40,43]. The interior strength of Didymos, however, must not be so large that it prevents the asteroid from flattening (unless the asteroid were formed in a flattened state to begin with). To permit such internal deformation while also shedding mass, asteroid spin-up calculations first used at Bennu[43], but adapted for Didymos, show that an interior average bulk $C$ of $\sim 10$ Pa (shear stress level at zero pressure) is required near the current spin-rate for a bulk density of 2800 kg/m³ and a friction angle of 35° (Fig. 9). The deformation would permit the observed flattening of Didymos and the non-circular equatorial circumference, as any interior mass concentration would shift outwards. Anything stronger than the $C$ of >20 Pa would prevent any interior deformation until Didymos spins up to <2.21 hours. Interior strengths similar to the surface $C \leq 1$ Pa would see the entire asteroid deform into a near pancake (which is not observed) before mass shedding begins. It should be noted that at the center of Didymos, the overburden stresses would exceed 100 Pa, and friction effects, rather than cohesion, dominate. The spin-up calculations also indicate that 10 m-radius boulders would need to travel more than 100 m along the surface to possess enough momentum to be ejected and enter orbit.

On Dimorphos, rocks and boulders sitting on larger boulders allow estimating strength properties of the unconsolidated surface of this asteroid. The analysis shows (Fig. 6d) that no rock or boulder sits on any slope exceeding 35°. In the case where no surface cohesion exists, this 35° angle must equal the greatest friction angle that can exist between rocks or boulders on Dimorphos. A separate effort that makes a detailed assessment of the shapes of the boulders on Dimorphos arrives at a near identical friction angle[44]. Nevertheless, since the friction angle has not been directly measured, we also

consider the possible presence of cohesion. As an upper estimate of its value, we assume the smallest reasonable static friction angle possible for natural materials to be $\sim 25°$ [45]. Again, using the same FS approach above, for a 5 m-thick debris layer on Dimorphos, which matches the heights of some the largest boulders on the asteroid, we estimate at most a $C \simeq 0.3$ Pa to maintain the rocks on 35° slopes. Consistent with this estimate of $C$, calculations which match the observed momentum transfer observed from DART[32], show that surface $C$ cannot exceed 0.9 Pa if it is a 15 m-thick layer, while overlaying a stronger interior of $Y > 500$ Pa.

It should be noted that $Y$ of individual boulders on Dimorphos likely significantly exceeds this 0.9 Pa upper limit for Dimorphos' surface. We find evidence for at least two small craters on boulders (Fig. 4b). A systematic analysis of craters on boulders can be used to estimate the strength of surface boulders[20] and such work is ongoing for Dimorphos. Both candidate boulder-craters possess a region that resembles spall surrounding their crater pit (Fig. 4b), not uncommonly

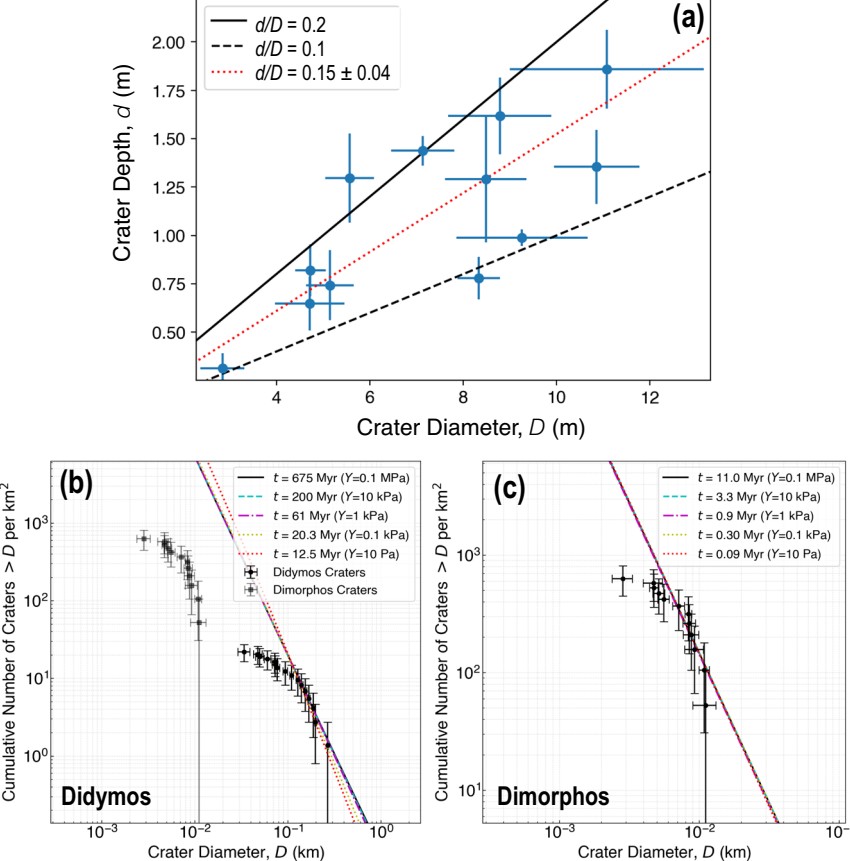

**Fig. 7 | Crater shapes and surface ages of Dimorphos, and surface ages of Didymos.** Geometric[29] crater depth to diameter ratio (**a**) measured on Dimorphos, and crater size frequency distribution on Didymos (**b**) and Dimorphos (**c**). Errors bars show 1-σ uncertainty. Dotted red-line shows (**a**) $d/D$ results for Dimorphos from DTMs using the latest GDTM[85]. Didymos' GDTM was of insufficient quality to measure reliable $d/D$.

observed for impacts into tough, non-porous rocks[46]. This likely indicates that the individual boulders on Dimorphos possess reasonable cratering strengths near several 10 s of MPa typical of ordinary chondrites (a factor of 10 less than the dynamic compressive strength of an ordinary chondrite[47]).

We suspect that $Y$ for Dimorphos may be between the strength holding the individual unconsolidated surface boulders together (<0.9 Pa) and the individual boulders comprising the surface (10 s of MPa). We interpret the aforementioned lineament fabric, long and narrow lineaments, and trough (Fig. 4d–f) as evidence that Dimorphos can sustain stresses over long (50–80 m), relatively straight distances via interior $Y$ in excess of 0.9 Pa. Studies have shown that damaged rocks, which likely make up Dimorphos, are weak relative to their original, non-damaged constituent rocks, but are strong enough to allow lineaments to form via impact[48], and can sustain fracturing from tidal stresses.

The geological evidence for Dimorphos, therefore, suggests that prior to DART's impact, it possessed a well-connected and/or well-packed set of fragments, below the unconsolidated, low $C$ (≤ 0.9 Pa) surface layer, whose bulk $Y$ likely exceeds the maximum surface $C$ of 0.9 Pa. A strength increase with depth is in line with interpretations obtained at Eros[49], Bennu[25,33], and Ryugu[36]. In the case of Eros (33 × 13 × 13 km), it has been proposed[49] that the surface strength envelope might change with depth because of a transition from a loose surface regolith to a coarse megaregolith similar to what is seen on the Moon; for Ryugu and Bennu a range of hypotheses have been proposed, including the presence of a layer of cohesive fines near 5–8 m depth[37,50]. In the case of Ryugu, this layer had a compressive strength of 140–670 Pa[34]. From telescopic

observations of the DART impact[51], there is some evidence for fines (<100 $\mu$m) produced in telescopic data in volumes that may exceed what could have just been produced from the crushing of Dimorphos surface rocks alone. Perhaps Dimorphos' apparent strength is the result of the presence of several large but damaged ordinary chondrite-like fragments that become increasingly packed at depth, with some cohesive glue created by fines.

Models of the DART impact indicate that the momentum change observed[50] would require an $Y$ anywhere between a few Pa to 10 kPa. A $Y$ below ~5 Pa implies cratering on Dimorphos is effectively in the gravity crater regime[51].

### Implications for Didymos and Dimorphos surface ages

The DART experiment[31,32] and asteroid near-surface strength assessments on other asteroids[33–36], provide an opportunity to connect surface strengths similar to those derived from observations of slope and numerical models to the $Y$ used in scaling relationships from which absolute surface ages of asteroids are determined. By matching simulated values of the momentum enhancement factor, $\beta$, with the outcome of the DART experiment[52], we[31,32] were able to make predictions for the strength of the Dimorphos surface and the size of the DART crater. Here, we input the known DART impact energy[1] and the predicted diameter range for the DART crater, $D_{DART}$ = 30, 60, and 80 m, into the crater-scaling relationships to determine corresponding cratering strengths for Dimorphos, $Y$ = 1300, 40, and 5 Pa, respectively. We note these values for $Y$ reflect input parameter values for the strength scaling relationships, rather than a physically measured value of the strength of Dimorphos at the DART impact site, and do not account for effects of surface curvature.

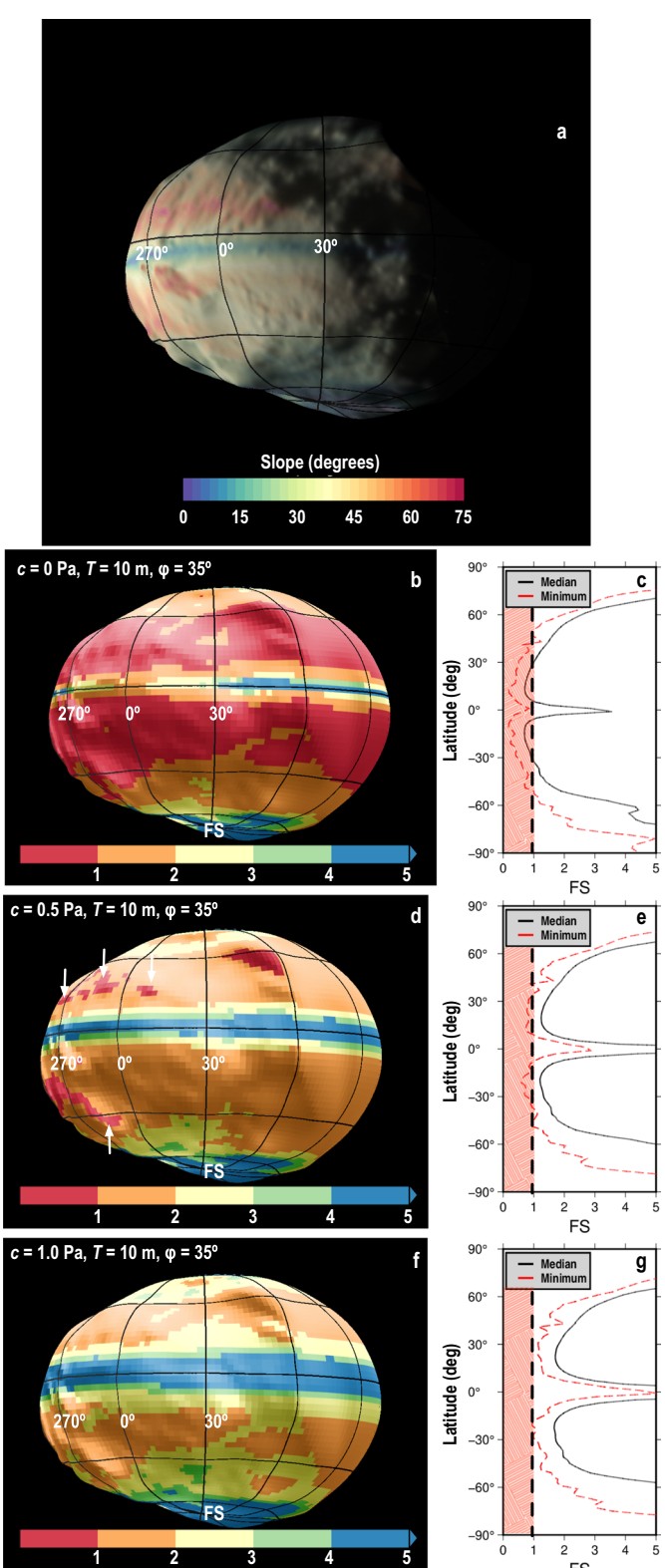

**Fig. 8 | Slope stability assessement of regolith on Didymos.** Didymos slope (**a**) and factor of safety (FS) for surface cohesion, $C = 0$ Pa (**b, c**), 0.5 Pa (**d, e**) and 1 Pa (**f, g**), assuming a 10 m-thick unconsolidated layer with friction angle $\phi = 35°$. Arrows in **d** show plausible boulder track and landslide sources.

The surface strengths inferred for both Didymos and Dimorphos indicate that smaller $Y$s are probably more reasonable. In the case of Didymos, we find a surface layer with $C$ of $\leq 2$ Pa covered by a slightly

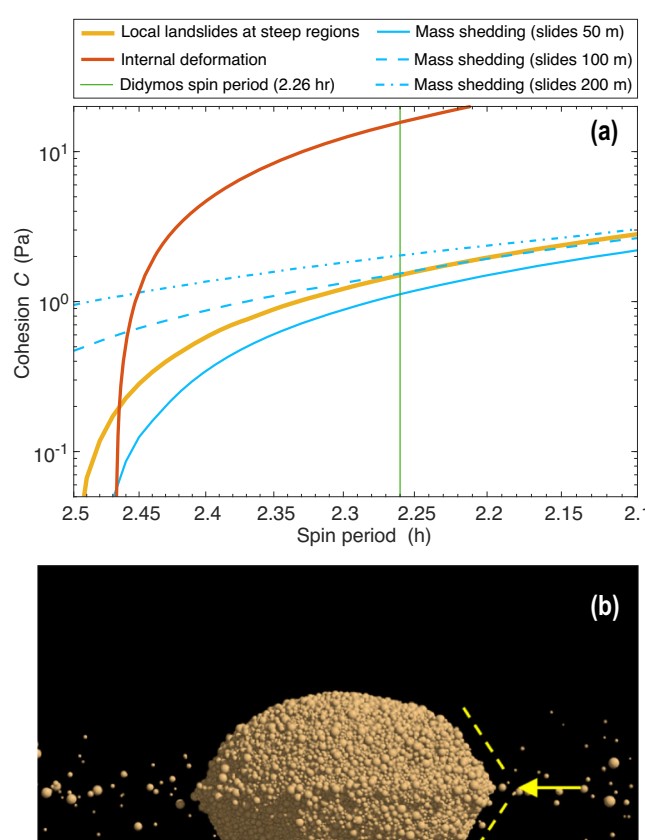

**Fig. 9 | Failure and reaccumulation of Didymos[38]. a** Failure-mode diagram of Didymos for a density of 2800 kg/m³. These curves mark the minimum cohesion required to prevent structural failure in Didymos across different spin periods. The analysis considers three potential failure modes common to rubble-pile objects. Each curve is derived based on the corresponding failure criterion. Interior deformation that affects the entire asteroid occurs if C is below the red curve, which for Didymos requires an interior bulk $C \lesssim 10$ Pa near its current spin rate of 2.26 h (vertical green line). Landsliding and mass-shedding (orange) can be initiated when the surface $C < 1$ Pa. Mass shedding (blue curves) is derived by placing a 10 m-radius boulder on the surface that slides over a given distance under the local gravity. For 1 Pa surface cohesion and the current spin rate of Didymos, a boulder would need to roll $\geq 100$ m before it lifted off the surface by the Coriolis force. **b** A soft-sphere discrete element simulation of a Didymos-shaped rubble pile shows that the mass movement and redeposition of shed material onto the equatorial region could help to form a triangular ridge on Didymos.

stronger interior near 10 Pa. The Didymos surface age estimates calculated with similar $Y$ (1 to 10 Pa) yield a surface age near 12.5 Myr.

In the case of Dimorphos, the craters are small ($\leq 10$ m) in diameter suggesting that the impact energy required to create them is not very high. The impactors for these craters probably never penetrated below a few meters in depth, and so are less likely to have encountered a stronger substrate inferred by the lineaments. The lower-end of strength (0–0.1 kPa) estimates determined by numerical simulations of the DART impact[31,32] and inferences from Ryugu are, therefore, preferred. Consequently, the surface age of Dimorphos must also be well-modeled by impacts in a weak (near the gravity regime) target, with a surface age of 0.09–0.3 Myr (Fig. 7c).

Due to the high boulder number density on the surface of Dimorphos, we cannot discount the possibility that impact armoring[53] played a role in reducing the number of observable craters on the hemisphere observed by DRACO. A reduction in the population of craters <2 m was reported on Bennu[28]. As the number density of boulders on Dimorphos is larger than on Bennu[11], we might expect that the transition into the armoring regime occurs at a larger diameter, maybe even at ~ 5 m where we see the break in slope of the Dimorphos CFSD (Fig. 7). But this transition could be offset by the greater yield strength of individual S-type boulders relative to more crushable C-type ones that may reduce the efficiency of armoring[54]. The exact consequence on the true surface age of Dimorphos is, therefore, not exactly known and is likely a second-order effect compared to the poorly understood cratering strength.

## Discussion

The DART mission has provided the first close-up views of a small NEA binary system, whose origins have been the subject of numerous theoretical and computational studies[55]. The leading hypothesis for the formation of NEA binaries is that radiative torques (i.e., the Yarkovsky–O'Keefe–Radzievskii–Paddack or YORP effect)[56] spin up a rubble pile primary to the point of failure, leading to a large mass shedding event that places material into orbit around the primary. This material accretes to form the secondary. In this scenario, both the primary and secondary are thought to be rubble piles with different formation ages.

In agreement with theoretical expectations, our geological assessments of Didymos are consistent with it being a rubble pile. Didymos has several large boulders with sizes too large to be excavated by the observed craters[57]. Rather, these boulders must have formed via catastrophic disruption of Didymos' parent body[58], which then gravitationally reaccumulated to form the largest components of a rubble pile. But the density of $2800 \pm 280$ kg/m³ for Didymos is large, implying a low total porosity near $21 \pm 8.5\%$ when compared to a typical L or LL chondrite with grain density of 3580 kg/m³ or 3520 kg/m³, respectively[59]. L or LL chondrites best match the spectral properties of the Didymos system[60]. We note that the estimated volume of Didymos is probably too small because the current GDTM does not match the non-illuminated limb (Fig. 3). The true porosity of Didymos, therefore, is likely greater than reported here. The relatively low porosity could be reconciled with a rubble pile structure if Didymos is made of several large aggregates and a thin ($\leq 10$ m), loose, and unconsolidated surface. The large interior aggregates would account for the non-circular equatorial shape. Alternatively, as proposed for Ryugu[61], the boulders comprising Didymos could be well-packed, maybe as a result of deformation, efficiently eliminating some of the porosity within it but allowing some large interior boulders to generate some of perturbations in its equatorial shape.

The observational evidence for mass motion and mass shedding on Didymos also supports theoretical expectations for the origin of Dimorphos. Several studies[40,43,55] show that a cohesive interior greatly facilitates surface shedding for asteroids that spin-up, indicating that some strength must exist at depth for Didymos. Figure 9a indicates that in order to have some shape deformation (to explain Didymos' squashed appearance if it were not born squashed) and mass mobilization at a spin rate marginally exceeding Didymos' current rotation, the overall average bulk $C$ of Didymos cannot exceed 20 Pa. The triangular shape of the Didymos ridge (Fig. 2a) may have been created by the mass shedding event (Fig. 9b).

Didymos' CSFD and our analysis of its strength properties indicate that its surface age is ~12.5 Myr. It is unlikely that a surface mobilization event that led to the formation of Dimorphos could have completely reset the surface age of Didymos. Numerical simulations[38] show that mass shedding events are relatively efficient at forming asteroid satellites, requiring only ~ 2–3% of the Didymos volume to be shed to form Dimorphos. Such a volume of material would be equivalent to a 3–5 m deep global layer (this layer would be deeper if mass was shed from a localized area). This is insufficiently thick to wipe away the largest craters on Didymos with depths near ~15 m; these largest craters define the age of Didymos. As a consequence, we would expect, if the bulk shape were maintained, that the largest craters on Didymos could be relatively well preserved following a mass shedding event. While we cannot rule out effects of image resolution, the difficulty in clearly identifying craters <100 m-diameter on Didymos may be the result of surface modification caused by the deformation, landsliding, and mass-shedding. Similarly, the largest craters could be relatively well preserved following a Didymos flattening event, if this event was modest[40].

We note that our estimated surface age for Didymos is substantially younger than the age determined for its most plausible asteroid family, the Baptistina family[62], which has an estimated age range of $140 - 320$ Myr[63]. Such an age would be better-matched if the surface strength of Didymos was ~10 kPa (Fig. 7), which may be plausible if the impacts that formed its craters interacted with a stronger sub-surface layer that is ~$15 - 30$ m deep. Evidence for a stronger sub-surface layer on other rubble pile NEAs has been noted before[22]. However, if the interior bulk cratering strength remains small near the 10-20 Pa value inferred by the possibility that Didymos became flattened, then Didymos may be an Nth-generation rubble pile[64], meaning that it did not directly originate from the catastrophic disruption of (298) Baptistina. Rather, Didymos may represent the latest of multiple asteroid generations stemming from the original parent[64].

Many attributes of Dimorphos' geology are in line with it being a rubble pile resulting from the gravitational accumulation of debris that was shed by Didymos. Similar to Didymos, no crater is large enough to explain the size of the largest boulder observed on this asteroid's surface, suggesting it inherited its constituent boulders from a larger parent body. Further, the lack of any preference in the total number or size of boulders with latitude and longitude, the presence of an intimate mixture of boulders and cobbles, and the lack of obvious fine regolith are all consistent with Dimorphos being the result of mass shedding from Didymos and gravitational re-accumulation. Material shed from Didymos is expected to form a ring of material that includes all sizes around Didymos[40,42,65], except for the very finest (<100 μm) particles. These finest particles get quickly swept away by solar radiation forces[66], within hours as exemplified by observation of the DART tail[51]. The rest of the rocks accumulate quickly, in a handful of days. Their random distribution, regardless of latitude or longitude (Fig. 5) is consistent with models of secondary formation using large discrete element particles[38] where the satellite often forms with a non-principal axis rotation, and can accrete material at all latitudes and longitudes. These large particle models, however, also usually generate an elongated secondary, which Dimorphos was not. Some additional accumulation process, therefore, may be required to explain its oblate shape.

That Dimorphos is brighter than average Didymos[12] could provide further evidence that Dimorphos recently accumulated from Didymos debris. Not only does Didymos possess at least one very bright spot near steep slopes, it is well known that less heavily space-weathered S-type material, often exposed on steep slopes, is brighter than weathered S-type material[67]. We hypothesize, therefore, that the aggregates of Dimorphos were originally refreshed possibly via surface erosion when shed, or derived from fresher material (e.g., the subsurface; steep regions). We further hypothesize that if mass shedding that formed Dimorphos is localized to mid-latitude regions, then, based on our arguments for the retention of large craters, Didymos craters $\lesssim$50-m in diameter would be depleted in those regions.

Dimorphos' CSFD provides our best estimate of the surface age of the satellite (~$0.09–0.3$ Myr), but the interpretation of this age is

complicated by several mechanisms that could plausibly reset it. Combined with the geophysical observations of Didymos outlined above, our analysis provides a handful of hypotheses on the geologic history of the binary system:

1. The surface age of Dimorphos is equivalent to its formation age. Didymos formed in near-Earth space, experiencing a large mass shedding event $0.09 - 0.3$ Myr, which also caused Didymos to flatten, but preserved its largest craters. The preferential distribution of boulders on both Dimorphos and Didymos provides insight into the reaccumulation dynamics of asteroid satellites, which gives a test for numerical models of asteroid satellite formation[38]. A depletion of small craters (< 100 m) is expected below steep slopes of Didymos.

2. The surface age of Dimorphos was reset by episodic mass exchange. Following a large mass shedding event that led to the formation of a proto-Dimorphos in orbit around Didymos, periodic failure of Didymos' surface led to mass exchange between the primary and the satellite, leading to satellite resurfacing. The last of such follow-on shedding events occurred $\sim 0.09 - 0.3$ Myr. Material preferentially accreted onto Dimorphos' equator, leading to a more oblate shape and erasing its past cratering record. This scenario provides a separate constraint for numerical models of binary asteroid evolution. Intermediate to small craters on Didymos should be present below some steep slopes, but heavily modified below others.

3. The surface age of Dimorphos was reset by impacts. Alternatively, external energy input from collisions destabilized the system $0.09 - 0.3$ Myr, leading to reshaping and resurfacing through impact-induced seismic shaking[68] or non-principal axis rotation[69]. Because previous impacts onto Dimorphos likely had energies much smaller than that provided by DART[62], measurements of the extent of surface modification following DART should constrain the level to which natural impacts could reset the surface of Dimorphos.

The Hera mission, developed by the European Space Agency, will be launched in October 2024 and will perform a 6-month exploration of the binary system in 2026[70], offering answers to some of the open questions left by our analysis. Better resolved, global imaging of Didymos, and a detailed analysis of Dimorphos should, with modeling, provide key tests for the hypotheses presented on age resetting mechanisms. Further, the low frequency radar onboard the Juventas cubesat will probe the interiors of Dimorphos and Didymos. These data will test the hypothesis that stronger, large blocks are present in their interiors, and will likely lead to additional discoveries on the physical attributes of both bodies. Complete image coverage of the surface of the two binary components at different wavelengths will improve current shape models, and provide additional geological and compositional constraints for understanding the origin and evolution of this system. Observations of the DART crater, supplemented by the landing of the CubeSats on Dimorphos and accompanying measurements of their bouncing properties, will reveal the surface strength properties of the asteroid. Hera, therefore, will ultimately check with a comprehensive, global dataset, many of the hypotheses and interpretations made here about the surface properties, geology, and history of the Didymos binary system derived from DART's small, restricted dataset. The result will not only provide a better understanding of the origin and evolution of the Didymos system, but also present an opportunity to understand how well a flyby/fly-in dataset can constrain the true nature of an asteroid.

## Methods
### Encounter conditions
The geometry of both the DRACO and LUKE data acquisitions limit the geological interpretations possible for both Didymos and Dimorphos.

DRACO images of Didymos and Dimorphos were all collected on a fixed intercept geometry near 60° solar phase, which is suitable for morphological assessments of their surfaces, but less suitable for interpreting their surface albedos. Both asteroids were viewed at a single solar illumination. The limited phase and illumination changes that are observed in the DRACO images are only due to the surface curvature of both asteroids.

In the case of Didymos, the DART spacecraft flew past the object, collecting images with all range of emission angles that provided some stereo parallax. LUKE collected images with phase angles ranging from 60–120°, as the LICIACube spacecraft flew past and observed the southern (-Z) hemispheres of the two asteroids ≈130 s after DART's impact[3,71]. About 40% of the total surface area of Didymos was observed with both cameras at a similar pixel scale of 5 m/pixel, with the best-resolved Didymos images collected by DRACO at ≈ 3 m/pixel.

In the case of Dimorphos (Supplementary Fig. 1), the impact of the DART spacecraft allowed for very limited stereo parallax[1]. In total, DRACO observed about 23% of the asteroid in solar illumination at a pixel scale of ≤ 0.29 m. The best-resolved, complete DRACO image was ≤ 0.055 m/pixel of a location surrounding the impact point. LUKE images of Dimorphos had fairly coarse pixel scales of 5.5 to 7 m/pixel and were heavily influenced by bright ejecta generated by the DART impact. As a consequence, they did not drive our geological interpretations of Dimorphos.

The images obtained by DRACO and LUKE were used to build global digital terrain models[1,10] of both asteroids using stereophotoclinometry (SPC)[72]. Because of the lack of global coverage, the SPC-derived models were further informed by observed limbs, and the location of the observed terminator[1]. Reflected light from Didymos and the ejecta produced by the DART impact itself revealed portions of the shapes of both asteroids not directly illuminated by the Sun (Fig. 3).

### Didymos digital terrain model
The Didymos GDTM was generated using stereophotoclinometry[72] combining images from both DRACO and LUKE. Although DRACO data were key to establish the initial scale of Didymos because the DART trajectory and pointing were better known than LICIACube's, the best and most consistent solutions for Didymos' shape were obtained by weighing equally DRACO and LUKE data within the SPC framework. The DRACO data provided the initial scale of Didymos and some stereo that constrains the visible equatorial perimeter of Didymos. These data allowed registering the LUKE data obtained in a similar geometry. Additional LUKE images whose geometry differed from DRACO (imaging mostly the south pole of the asteroid) were then added to the system to complete the process of generating a GDTM and producing a well registered set of LUKE images. Note that DRACO limb information was not used in building the Didymos GDTM presented, which explains why the triangular shape of the ridge and boulder profiles visible in the limb of Fig. 2b were not captured in the shape (Fig. 2 d and e). The original GDTM produced did have a center-of-mass (COM) to center-of-figure (COF) offset of (16, 42, −33) m. Given the uncertainties in spacecraft state (position and attitude) associated with the data that went into the model, we did not put much stake in this offset, and re-centered this COM-COF to zero. This re-centering of the Didymos GDTM was further justified given evidence that the unlit portions of the asteroid underestimated the extent of the asteroid limbs seen in back-lit images (Fig. 3b). The consequence of this underestimate is that the Didymos GDTM is slightly too small (in the $a - b$ plane). The GDTM of Didymos produced here is 11% smaller in volume relative to the prior radar model, just within the 12% uncertainty (1-$\sigma$) associated with the radar model[4]. Supplemental Table M1 provides additional characteristics of the model presented. The uncertainties presented are the result of analyzes[1,10,73–75], using limb and terminator assessments and surface feature location differences seen in the original images and renderings of the shape model.

## Definition of surface elevation

Surface elevation (Fig. 2d) is computed as the difference between the gravitational surface potential and a reference potential divided by the magnitude of the local gravitational acceleration, $g$, considering reasonable estimates of the asteroid's bulk density (2800 ± 280 kg/m³; [31,76]) and spin period (2.26 h[77]) at the time of writing. The uncertainty in bulk density will be resolved by the Hera mission and arises because of remaining uncertainty in the size of Didymos[76].

## Didymos crater identification

The identification of Didymos candidate craters was challenging due to two factors: 1) the fairly low imaging resolution available from both DART and LICIACube (Supplementary Fig. 1), and 2) the apparent ongoing surface mass motion that is probably a consequence of Didymos' rapid rotation (2.26 hrs). In order to develop a plausible list of candidate craters, several of us independently assessed all the available images for Didymos with the aid of a Didymos GDTM. Although the GDTM employed was not good for measuring depths of plausible crater candidates, the model did provide key qualitative data on whether or not the candidate craters had a circular appearance, were deeper relative to their surroundings, or had other attributes associated with craters at other asteroids. The craters were identified using the images and GDTM and a set of observational criteria listed in Supplemental Table 2. These were intended to provide an objective means for establishing confidence in the craters identified, which totaled 16 (Supplemental Table 3).

An image of each crater is shown in Supplemental Material Table 3, along with its location and confidence rating.

## Dimorphos crater identification and morphological assessment

We identified, mapped, and assessed crater candidates on Dimorphos through a series of three steps: 1) initial mapping, 2) a triage step using the global DTM, 3) morphological assessments using local DTMs of candidate craters that passed step 2.

**Step 1.** Craters were identified in DRACO images of Dimorphos by projecting images onto the Dimorphos v003 shape model[1] using the SBMT[26]. The pixel scales of the images used to map craters are between 0.05 m/px and 0.23 m/px. Our initial search yielded 22 mapped candidate craters. Mapped craters had diameters that ranged from 2.9 to 11.1 m and were identified as circular features composed of smaller interior particles encircled by larger cobbles or boulders. These crater characteristics are similar to those of small (<10 m) craters on the rough surfaces of Bennu and Ryugu[25,28,78].

**Step 2.** Using the highest resolution Dimorphos v003 shape model (average facet area of 0.024 m²), three linear cross-cutting profiles were mapped onto each of the 22 candidate craters. The mapped profiles provided the variation in height as a function of distance across each crater. These data were inspected to determine if any of the three profiles had a classical bowl shape or the presence of a central depression. Of the 22 candidate craters, 15 had at least one bowl-shaped profile.

**Step 3.** Local DTMs were constructed for the 15 candidate craters which passed the triage step. Using the DTM, the crater diameter, $D$, and depth, $d$, were measured using 8 profiles that cross-cut the center of the crater. The geometric height was used to measure the depth-to-diameter ratio of each crater. Of the 15 candidate craters, 12 were verified to be craters through this morphological assessment. These are shown in Supplemental Table 4 along with additional information on their diameter and depth.

## Assumed impactor populations for calculating surface age

The surface ages presented here take into account the impact history of each body due to collisions with other smaller asteroids. Didymos likely originated in the main asteroid belt (MBA), where the population of impactors is 2–3 orders of magnitude greater than that in near-Earth space. As recorded by the CSFD of its largest craters, the surface age of Didymos most likely reflects that MBA impactor population[79] before Didymos became a near-Earth asteroid. We use, therefore, an MBA impactor population distribution with intrinsic collisional probability, $<P_i> = 2.86 \times 10^{-18}$ km⁻² yr⁻¹, and a fixed average representative impact velocity, $U = 5.6$ km s⁻¹ [80]. We also note that the CSFD of craters on Didymos with a diameter >100 m follow reasonably well the MBA distribution with a $-2.4 \pm 0.2$ slope in a log-log plot. Following the above assumptions, and considering a bulk density of 2800 kg m⁻³, we calculate that Didymos' surface age ranges between 12.5 and 675 Myr (Fig. 7).

In Dimorphos' case, we consider a mixed population of near-Earth asteroids (NEA) and MBA impactors[49], given its residence in the inner main belt (semimajor axis > 2.1 AU) for approximately a third of its current orbit. Our assumption is that the small CSFD observed on Dimorphos must have occurred recently and may reflect the time since Didymos rotationally disrupted through radiative torques (i.e., the YORP effect, see the Discussion section). Despite spending the majority of its orbit outside the main belt, the impact flux onto Dimorphos is still dominated by MBAs. Furthermore, NEAs with orbits that cross into the inner main belt can experience larger average impact speeds and impact probabilities[81], and an asteroid with Dimorphos' orbital parameters would thus have intrinsic collisional probability, $<P_i> = 11.2 \times 10^{-18}$ km⁻² yr⁻¹, and a fixed average representative impact velocity, $U = 7.1$ km s⁻¹. We find that Dimorphos' crater CSFD has a slope of $-2.4 \pm 0.5$ for craters >5 m-diameter, which falls within that of the MBA distribution, though the uncertainty is quite large due to small number statistics. Following the above assumptions, and considering a bulk density of 2800 kg m⁻³ [1], we calculate that Dimorphos' surface age ranges between 0.09 and 11 Myr for the same $Y$ between 10 and $10^5$ Pa, respectively (Fig. 7). The surface age values we calculate, especially for weak surfaces, are much smaller than the dynamical lifetimes of NEAs[82], which agrees with our initial choice of a mixed impactor-flux model for Dimorphos.

## Data availability

The DART mission archive at NASA's Planetary Data System contains data from DRACO, as well as associated documentation and advanced products, including the shape models of Didymos and Dimorphos (https://pds-smallbodies.astro.umd.edu/data_sb/missions/dart/index.shtml and https://naif.jpl.nasa.gov/pub/naif/pds/pds4/dart/dart_spice/). We use the latest models of Didymos (v003) and Dimorphos (v004). The Small Body Mapping Tool (SBMT; https://sbmt.jhuapl.edu/) developed by Johns Hopkins Applied Physics Laboratory contains the shape models of both asteroids as well. These can be viewed in conjunction with the DRACO images. Slope and elevation data shown in Figs. 2 and 6 have been modified from what is in the PDS and are publicly available in the SBMT. All the source data for the graphs presented are available in a source data file. Source data are provided with this paper.

## Code availability

The simulation shown in Fig. 9(b) was performed by the code PKDGRAV in its custom version that includes the Soft-Sphere Discrete Element Method. Several compiled versions of PKDGRAV are provided as Supplementary Software 1. Visualization support of Fig. 9(b) is provided by the open-source POV-Ray ray-tracing package (https://www.povray.org).

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

## Acknowledgements

This work was supported by the DART mission, NASA Contract 80MSFC20D0004, the Italian Space Agency (ASI) within the LICIACube project (ASI-INAF agreement n. 2019-31-HH.0) and the Hera project (ASI-INAF agreement n. 2022-8-HH.0). N.M. and C.R. acknowledge funding support from the European Commission's Horizon 2020 research and innovation programme under grant agreement No 870376 (NEO-MAPP project) and support from the Centre National d'Etudes Spatiales (CNES), focused on the Hera space mission. SDR acknowledges support from the Swiss National Science Foundation (project number 200021_207359). R.N. acknowledges support from NASA/FINESST (NNH20ZDA001N). TK is supported by the Academy of Finland project 335595 and by institutional support RVO 67985831 of the Institute of Geology of the Czech Academy of Sciences. JMT-R acknowledges support from the Spanish project PID2021-128062NB-I00 funded by MCIN/AEI.

## Author contributions

O.B. led the presented research effort presented and is responsible for the final manuscript. R.-L.B. undertook most of the crater measurements presented. S.M., R.-L.B., and K.W. led the surface dating efforts and provided key interpretations of findings. J.-B. V. provided roughness assessments. J.-B. V. and M.P. verified craters identified on Didymos.

H.A., Y. Z., P.M., F. F., M.H., D.C.R., and O.K. provided key insights on the plausible formation and evolution mechanism of the Didymos system. M.P., F.T., and A.L. undertook the boulder counts and surface lineament assessments presented. O.B., C.M.E., T.D., T.F., and E.P. were instrumental in generating the GDTMs of Dimorphos and Didymos. J.M.S., J.L.R., P.H.A.H., L.P., N.M., C.Q.R., T.K., E.A., M.N., A.C.B., and J.M.R. provided key albedo assessments and insights on the geological processes observed. S.R., M.J., and A.S. provided key impact cratering findings. J.-Y. L. provided key thoughts on the evolution of small particles. N.C., A.S.R., and A.F.C. made the DART investigation possible. E.D., V.D.C., E.E., I.G., J.D., I.B., A.Z., J.B., S.L.I., J.R.B., G.P., G.Z., M.A., A.C., G.C., M.D., S.I., G.I., M.L., D.M., P.P. D.P., S.P., P.T., M.Z. made the LICIAcube mission possible, and the LICIAcube images investigated here.

## Competing interests

The authors declare no competing interests.

## Additional information

Olivier Barnouin [1] ✉, Ronald-Louis Ballouz [1], Simone Marchi [2], Jean-Baptiste Vincent [3], Harrison Agrusa [4,5], Yun Zhang [6], Carolyn M. Ernst [1], Maurizio Pajola [7], Filippo Tusberti [7], Alice Lucchetti [7], R. Terik Daly [1], Eric Palmer [8], Kevin J. Walsh [2], Patrick Michel [4,9], Jessica M. Sunshine [5], Juan L. Rizos [10], Tony L. Farnham [5], Derek C. Richardson [5], Laura M. Parro [11], Naomi Murdoch [12], Colas Q. Robin [12], Masatoshi Hirabayashi [13], Tomas Kahout [14], Erik Asphaug [15], Sabina D. Raducan [16], Martin Jutzi [16], Fabio Ferrari [17], Pedro Henrique Aragao Hasselmann [18], Adriano CampoBagatin [11], Nancy L. Chabot [1], Jian-Yang Li [8], Andrew F. Cheng [1], Michael C. Nolan [15], Angela M. Stickle [1], Ozgur Karatekin [19], Elisabetta Dotto [18], Vincenzo Della Corte [20], Elena Mazzotta Epifani [18], Alessandro Rossi [21], Igor Gai [22], Jasinghege Don Prasanna Deshapriya [18], Ivano Bertini [23], Angelo Zinzi [24], Josep M. Trigo-Rodriguez [25], Joel Beccarelli [26], Stavro Lambrov Ivanovski [27], John Robert Brucato [28], Giovanni Poggiali [28], Giovanni Zanotti [16], Marilena Amoroso [24], Andrea Capannolo [17], Gabriele Cremonese [7], Massimo Dall'Ora [29], Simone Ieva [18], Gabriele Impresario [24], Michèle Lavagn [17], Dario Modenini [22], Pasquale Palumbo [23], Davide Perna [18], Simone Pirrotta [24], Paolo Tortora [22], Marco Zannoni [22] & Andrew S. Rivkin [1]

[1]Johns Hopkins University Applied Physics Laboratory, Laurel, MD, USA. [2]Southwest Research Institute, Boulder, CO, USA. [3]DLR Institute of Planetary Research, Berlin, Germany. [4]University of the Côte d'Azur, Observatory of the Côte d'Azur, CNRS, Laboratory Lagrange, Nice, France. [5]University of Maryland, College Park, MD, USA. [6]University of Michigan, Ann Arbor, MI, USA. [7]INAF-Astronomical Observatory of Padova, Padova, Italy. [8]Planetary Science Institute, Tucson, AZ, USA. [9]The University of Tokyo, Department of Systems Innovation, School of Engineering, Tokyo, Japan. [10]Institute of Astrophysics of Andalusia, CSIC, Granada, Spain. [11]University of Alicante, Alicante, Spain. [12]Superior Institute of Aeronautics and Space, University of Toulouse, Toulouse, France. [13]Daniel Guggenheim School of Aerospace Engineering, Georgia Institute of Technology, Atlanta, GA, USA. [14]Univ. of Helsinki, Helsinki, Finland. [15]Lunar and Planetary Laboratory, University of Arizona, Tuscon, AZ, USA. [16]Univ. of Bern, Bern, Switzerland. [17]Politechnic University of Milan, Milan, Italy. [18]INAF-Astronomical Observatory of Rome, Rome, Italy. [19]Royal Observatory of Belgium, Brussels, Belgium. [20]INAF-Institue of Space Astrophysics and Planetology, Rome, Italy. [21]Institue of Applied Physics "Nello Carrara", CNR, Florence, Italy. [22]University of Bologna, Bologna, Italy. [23]University of Parthenope, Parthenope, Italy. [24]ASI, Rome, Italy. [25]Institute of Space Sciences (CSIC-IEEC), Barcelona, Catalonia, Spain. [26]Univ. of Padova, Padova, Italy. [27]INAF- Astronomical Observatory of Trieste, Trieste, Italy. [28]INAF- Astronomical Observatory of Arcetri, Arcetri, Italy. [29]INAF- Astronomical Observatory of Capodimonte, Capodimonte, Italy. ✉e-mail: olivier.barnouin@jhuapl.edu

