## [Peer Review File · Nature Communications]

The geology and evolution of the Near-Earth binary asteroid system (65803) DidymosREVIEWER COMMENTS

Reviewer #1 (Remarks to the Author):

The authors analyze disk-resolved images of Didymos binary system collected by the DART and LICIACube spacecraft in order to access its geology and evolution. In general, the study is well-presented and scientifically relevant.

I provide some concerns below that might be relevant or could stem from my limited understanding of the subtopic, thus misunderstanding.

I have issues with several claims in the abstract.

First, "Didymos has a non-circular equatorial perimeter ". This is inferred from image 2a, if I understand that correctly. However, given the orientation indicated by the axes, the viewing geometry is far from being pole-on, perhaps by ~ 30 degrees. If so, then any interpretation concerning the equatorial perimeter is invalid. Also, part of the projected disk is not illuminated, about half of the contour is thus not defined. Some parts of the work use this claim as input for further interpretation. I am not sure if these interpretations are valid as well.

Second, "At high elevations, its surface is undulating and contains large boulders and craters; at low elevations, its surface is smoother and possesses fewer large boulders and craters." No boulder counts and statistics on Didymos are provided, so this is not supported by the data.

Third, "Dimorphos possesses a uniform surface covered by boulders." This is probably caused by inconvenient wording. This statement can also be understood in the sense that boulders are distributed uniformly in size, which is not correct according to Fig. 5. Moreover, I have some concerns about the analysis of the boulder statistics. The larger boulders (orange) are much more uniform than the smaller ones. Also, the histogram is not symmetric wrt the equator, but rather wrt the sub-spacecraft point. I wonder if viewing effects could corrupt/bias the boulder counts for larger observing latitudes. The histogram looks suspiciously similar to the cosine function.

Finally, "If part of the Bauphistina family, Didymos likely represents the latest of multiple asteroid generations stemming from the original parent." This scenario is presented in the text as an alternative to a different scenario. Both scenarios are still valid and the selection of just one for the abstract is incorrect.

Line 22 and Fig. 2a: The triangular shape is a local feature as it is not visible in Fig. 2d. Also, such a feature is prone to viewing geometry (it is not equator-on). It would be useful to indicate the orientation by the axes as it is done in Fig. 2b (instead of the arrow $z+$ which is not sufficiently indicative).

Fig. 2c: Three types of terrains. Could illumination/viewing geometry affect what you see? The surface seems similar if going from north to south (or top to bottom) along the terminator. Clearly, the three surface types are not preserved to the right (along the terminator), where shadowing reveals the roughness.

I have downloaded the global digital terrain model of Dimorphos from <https://lib.jhuapl.edu/papers/dart-an-autonomous-kinetic-impact-into-a-near-earth/> and I cannot match your figure 4f. I see a surface full of crater-like features in Fig. 4f, which is surprising and looks unrealistic given the disk-resolved images (and the model I have downloaded).

Line 317: Do we need a mechanism to flatten the shape of the primary? Could be already primordial, right? Asteroids are not born spheres.

Line 320: It is a speculation that is not addressed in the previous text.

Line 324: it is not clear to me what "are likely >15 -m-diameter " means.

I found some typos in lines: 19, 50 (Fig. 2a?), 54 (missing comma?), 61 (missing space between value and unit, multiple times in the text), 155 (input), 267.

Inconsistency between near-Earth and Near-Earth.

Weird numbering of sections (Acknowledgements and further).

Reviewer #2 (Remarks to the Author):

Referee comments on “The geology and evolution of a the Near-Earth binary asteroid system (65803) Didymos” written by Olivier Barnouin and others

This paper describes the first detail results of the geology on Didymos and Dimorphos studied by the images taken from DART and LICIACube spacecraft. The authors studied the shape and surface geology of both asteroids and determined the crater size frequency distributions and the boulder surface distributions. They also conducted numerical studies for the factor of safety on Didymos, and found that the surface cohesion could be less than 1 Pa to reproduce the boulder tracks found on Didymos. Furthermore, according to the previous studies for the shape of asteroid Bennu, they suggested that the interior cohesion of ~ 10 Pa was necessary for surface sliding, mass shedding, and some interior deformation. Similar to the discussion of Didymos, they also estimated the surface cohesion of Dimorphos and it was less than 0.9 Pa. Based on the surface cohesion of both bodies, the authors applied a crater scaling relationship for a crater chronology on both asteroids. Then, they estimated the surface age of both asteroids from the crater size frequency distribution. Finally, they found that the surface ages of Didymos and Dimorphos were very young compared to the Baptistina family, and they speculated that this means that the both asteroids could be the latest of multiple asteroid generations stemming from the original planet.

This paper is quite worthwhile for the advance of the study of rubble-pile bodies and binary-asteroids although most of the results may be confirmed or improved by the related mission of HERA, if it is succeeded. Moreover, this paper describes the geological features on both bodies before DART impact, so we can obtain quite important information by comparing pre- and post-impact geology on these bodies after HERA mission is completed. I think that this paper newly clarified other aspects of rubble bodies compared to Bennu, Ryugu, and Itokawa. Therefore, I strongly recommend the editor to publish this paper without delay. My major and minor concerns to this paper are described as follows. I appreciate if the authors answer to these comments.

Major comments:

One of the most important results of this paper is that the authors estimated the surface cohesion of the two rubble-pile bodies, and they are very small < 1 Pa. So, I think that Fig. 8 and Fig. 9 are important figures to determine them in this paper. Especially, the numerical calculation of Fig. 8 is very impressive for me to predict the boulder tracks. But there are some assumptions such as density, friction angle, upper layer thickness and rotation rate in this calculation. I am wondering how the FS is sensitive to these parameters, e.g. how much dose the FS change when the density changes from 3080 to 2520 kg/m³ or the friction angle changes from 20 to 40 degree? I think that this is closely related to the estimation of the surface cohesion. In addition to Fig. 8, Fig. 9 is also impressive for me. But I cannot follow how to derive several curves on Fig. 9 (a). Ref. 41 is cited

in this part, but I don't understand why 10 Pa cohesion is necessary in this model.

Minor comments:

1. Figure 3: There is no explanation for Fig. 3(a). Only (b) is described.
2. Line 42: "gsd" is not so common word. Please explain what "gsd" is.
3. Line 47-48: The following sentence is obscure, "The visual identification ... on both asteroids". Could you please rewrite clearly?
4. Line 68-71 and Fig.5: These sentences indicate that the boulder number density decreases with the increase of the latitude. I am wondering what materials occupy the inter space among boulders at high latitude. Is the space among boulders occupied by grains smaller than 0.5m?
5. Fig.6: I cannot imagine how to prepare Fig.6(c). What is the definition of slope? Is it an average of some area?
6. Line 89: Could you please explain "local debris aprons"? I don't see any apron in the photo.
7. Figure 7(a): What is μ in the caption? There is no definition of μ .
8. Line 155: "nputs" may be typo. It may be "inputs".
9. Figure 8: There is no number "a" in the top figure of Fig.8. The caption "C=0.5 Pa" of Fig.8f may be "C=1 Pa".
10. Line 177-178: What are the mass concentrations which induce the protrude? I cannot image it. Is it large blocks without porosity?
11. Line 184-185: Theses sentences may have a contradiction because it says that the strength is needed for mass-movements and mass shedding but the interior deformation is needed for the flattening. Could you please explain it more easily?
12. Line 196-198: I cannot understand the following sentences, "The deformation calculations ... and enter orbit". Please make clear. What is the deformation calculation and so on?
13. Figure 9(b): This figure is so impressive to explain the assumption that produce the triangular ridge and mass shedding. I recommend the authors to discuss this figure in the main text.
14. Line 234: "low C (le 0.9 Pa)" may be typo. It may be "low C (<0.9 Pa)".
15. Line 274: What is the origin of the width of a surface age, 0.09-0.3Myr? Do you use the different strength to obtain 0.09 and 0.3 Myr?
16. Line 280: Why does the author adopt 5m for the limit of armoring effect?
17. Line 394-395: The following sentence is obscure, so please rewrite it clearly, "Intermediate to small craters ... below others". What is "others"?
18. Line 417: "global datasaet" may be typo. It may be "global dataset"

emd

Reviewer #3 (Remarks to the Author):

This is an interesting paper detailing primary characteristics of the Didymos-Dimorphos binary system prior to the DART impact. Results here will be of broad relevance to NEO and asteroid studies, in particular, estimates of material strength and surface/formation ages here provide key constraints on system origin mechanisms and the overall (extremely young) system age. Methods and approaches utilized are generally state-of-the-art and well explained. I recommend its publication after addressing the minor issues indicated below.

lines 17-18 For the non-specialists, please clarify the definition of intermediate, major, minor, and z axes. It is notable that both objects have equatorial circumferences that are much closer to circular than most small bodies, i.e., they are much less ellipsoidal and are closer to oblate spheroids with a b/a near unity. This seems inconsistent with the tone of the first sentence in the abstract that instead emphasizes the equator of Didymos as being non-circular, when it instead appears to be remarkably near-circular.

line 17 Text is missing in parentheses after Fig. 1 reference

Fig. 1 caption: Are the terms "flattened" and "oblate" meant to convey different things? If yes, please clarify, and if not, use one term rather than both. Dimorphos seems similarly flattened, i.e., it has a similar c/a , to the secondaries in the plot, which seems to contradict the statement in the caption. What is notable is its high b/a , i.e., its more circular equator compared to other secondaries.

Fig. 3 caption: Please insert a size scale bar. What is the white region on the right side of (a)?
Line 61-63: 100-6 is an odd description; does this mean 6 to 100 cm in length? Please justify why a 16 m boulder is "too big to have been formed by local impacts".

Line 155: nputs \diamond input

Line 161: This is a surprisingly high density for a rubble pile, please comment.

Line 175 "non-circular perimeter at Didymos": Didymos is quite nearly circular at its equator ($b/a = 0.96$). Can the needed strength to account for $b/a = 0.96$ be quantified, and is this consistent with the estimates based on surface sliding and mass shedding cited on line 190?

Line 205: an near \diamond a near

Line 293-294: Is it possible to use the inferred age differences to place any meaningful limit on the radiation torques that led to the spin up of the primary?

Line 364-365: The accretion time of boulders in orbit around Didymos onto Dimorphos would seem to be extremely short. Is the tidal locking timescale short enough to explain why boulders are enhanced at low latitudes on Dimorphos, as suggested here? Or is it necessary to assume multiple mass shedding events from Didymos, with tidal locking occurring on much longer timescales between such events, as is described in hypothesis # 2 in lines 389-395?

Please find below our comments to three reviewers. This paper was greatly improved by their comments and we really appreciate their helpful inputs. Our responses are in blue. Text that are blue and italic show the updates that we made to the text. We have also provided a revised manuscript, plus a manuscript showing our changes relative to the original manuscript. Many of the changes made that do not directly address the reviewers' comments were undertaken because we identified additional typos, and improved the text for clarity while replying to the reviewers.

REVIEWER COMMENTS

Reviewer #1 (Remarks to the Author):

The authors analyze disk-resolved images of Didymos binary system collected by the DART and LICIACube spacecraft in order to access its geology and evolution. In general, the study is well-presented and scientifically relevant.

I provide some concerns below that might be relevant or could stem from my limited understanding of the subtopic, thus misunderstanding.

I have issues with several claims in the abstract.

First, "Didymos has a non-circular equatorial perimeter ". This is inferred from image 2a, if I understand that correctly. However, given the orientation indicated by the axes, the viewing geometry is far from being pole-on, perhaps by ~ 30 degrees. If so, then any interpretation concerning the equatorial perimeter is invalid. Also, part of the projected disk is not illuminated, about half of the contour is thus not defined. Some parts of the work use this claim as input for further interpretation. I am not sure if these interpretations are valid as well.

Response: Thanks for this comment, which must refer to original Figure 2b (not a). We have modified Figure 2 to clarify our observation and avoid any confusion. Although we have kept the arrows which show interpreted shape corners in Figure 2b, we removed the hashed circles in Figure 2b and any references to the off-axis observation. Instead, we have added a circular dotted line to Figure 2f, where we are looking directly at Didymos' GDTM (shape model) from the -Z pole. Both 2b and 2e images continue to show that Didymos is not circular (like Bennu and unlike Ryugu); but has plausible corners or edges like Bennu (due to its longitudinal, N-S ridges). The Didymos model is not sufficiently good to do a spherical harmonics assessment as was done for Bennu, to quantitatively confirm their presence, so this is best we can do until Hera arrives. Because these features are plausible, we believe the interpretation is also plausible and worth mentioning.

We have edited the main text also. See lines 22-26:

Unlike Ryugu or KW4 [5, and reference therein], Didymos' perimeter when viewed somewhat obliquely (Fig. 2b) or along its south pole (Fig. 2f), is not particularly circular, a trait that is reminiscent of Bennu, which has longitudinal (N-S) ridges that are apparent when Bennu is viewed from the same direction [6].

Second, "At high elevations, its surface is undulating and contains large boulders and craters; at low elevations, its surface is smoother and possesses fewer large boulders and craters." No boulder counts and statistics on Didymos are provided, so this is not supported by the data.

Response: Thanks for this comment as well: We forgot to add references to the submitted Pajola et al. (2023), which is now also added (line 39 – to be revised for the same special issue in *Nature Comms* that we submit this paper to; they provide their boulder data there). Their findings (see figure above from their manuscript) are consistent with the visual inspection provided here that we felt were fairly evident that regions at higher elevation possess more boulders than lows. There is an obvious band of smoothness at the low equator with no boulders visible in the Didymos images that is lacking at the high poles (where many large boulders are visible). The roughness assessments by Vincent et al. (2023; also submitted to the *Nature Comms* special issue), which were indicated in the original manuscript (line 50), also indicate roughness correspondence with elevation.

Third, "Dimorphos possesses a uniform surface covered by boulders." This is probably caused by inconvenient wording. This statement can also be understood in the sense that boulders are distributed uniformly in size, which is not correct according to Fig. 5. Moreover, I have some concerns about the analysis of the boulder statistics. The larger boulders (orange) are much more uniform than the smaller ones. Also, the histogram is not symmetric wrt the equator, but rather wrt the sub-spacecraft point. I wonder if viewing effects could corrupt/bias the boulder counts for larger observing latitudes. The histogram looks suspiciously similar to the cosine function.

Response: We have edited the wording "Dimorphos possesses a uniform surface covered by boulders" in the abstract with "Dimorphos possesses an intimate mixture of boulders". Since

submitting this paper, Pajola et al 2023 finalized his submitted paper on the boulder size frequency distribution and noted that boulders about $>2\text{m}$ are complete in the region mapped using a statistical methodology (see figure below). For this size of boulder, we stop seeing any latitudinal dependency on the size of boulders. We have updated Fig. 5 to reflect this change and have updated the wording in the text (see lines 68-71). We also indicate in the discussion, that this distribution of boulders better satisfies what would be expected for the formation of a secondary (line 370-374): *Their random distribution, regardless of latitude or longitude (Fig. 5), is consistent with models of secondary formation using large discrete element particles [40] where the satellite often forms with a non-principal axis rotation, and can accrete material at all latitudes and longitudes.*

Figure showing boulder size frequency distribution on Dimorphos. X_{min} provides a statistical estimate of the size of boulder above which the population is considered complete using a methodology described in DeSouza, I., et al. 2015. Improved techniques for size-frequency distribution analysis in the planetary sciences: Application to blocks on 25143 Itokawa. *Icarus* 247, 77–80. <https://doi.org/10.1016/j.icarus.2014.10.009>

Finally, "If part of the Baupristina family, Didymos likely represents the latest of multiple asteroid generations stemming from the original parent." This scenario is presented in the text as an alternative to a different scenario. Both scenarios are still valid and the selection of just one for the abstract is incorrect.

Response: Agreed. But this latter possibility is the most intriguing and plausible given the inferred weak material strength of the asteroid. We have removed this sentence in the abstract, but clarify why we think this is a very likely possibility in the main text. See new text at line 354-358.

However, if the interior bulk cratering strength remains small, near the 10-20 Pa value inferred by the possibility that Didymos flattened, Didymos may be an Nth-generation rubble pile [68], meaning that it did not directly originate from the catastrophic disruption of (298) Baptistina. Rather, Didymos may represent the latest of multiple asteroid generations stemming from the original parent [68].

Line 22 and Fig. 2a: The triangular shape is a local feature as it is not visible in Fig. 2d. Also, such a feature is prone to viewing geometry (it is not equator-on). It would be useful to indicate the orientation by the axes as it is done in Fig. 2b (instead of the arrow z^+ which is not sufficiently indicative).

Response: We have addressed this in the methods section now (line 656-682). The limbs from the DRACO images are not included in the low fidelity Didymos shape model; this results in a lack of triangular shape in the GDTM as seen in Fig. 2d and e. The images do show this triangular shape (Fig. 2a-c) and should be trusted; they are the real data. This addition of the limb data would likely enhance the square appearance of Didymos when viewed from -Z pole.

Fig. 2c: Three types of terrains. Could illumination/viewing geometry affect what you see? The surface seems similar if going from north to south (or top to bottom) along the terminator. Clearly, the three surface types are not preserved to the right (along the terminator), where shadowing reveals the roughness.

Response: The three terrain types do not extend towards the terminator as shown in Fig. 2c (now updated), but we do not think that it is a viewing issue. The incidence angle for much of the visible portion of the asteroid seen by DRACO is at very suitable angles for identifying morphology (see Method 1).

I have downloaded the global digital terrain model of Dimorphos from <https://lib.jhuapl.edu/papers/dart-an-autonomous-kinetic-impact-into-a-near-earth/> and I cannot match your figure 4f. I see a surface full of crater-like features in Fig. 4f, which is surprising and looks unrealistic given the disk-resolved images (and the model I have downloaded).

Response: The shape model you have downloaded is not the latest. The shape model used here is now available via the SBMT and the PDS (see data availability section). This model is also explained in detail in a separate paper now accepted at PSJ (Daly et al. 2024). This new model shows the distinct trough depression identified in the Fig 4f. It is resolved through photogrammetry in the disk-resolved images, where variations in brightness illuminate the trough. If you look closely at the images the troughs and lineaments are visible as well. We have now added a star indicating the location of DART's impact point, so it is easier for a reader to link the image Fig. 4e with the GDTM view shown in Fig. 4f.

Line 317: Do we need a mechanism to flatten the shape of the primary? Could be already primordial, right? Asteroids are not born spheres.

Response: Agreed. And we have made it clear in earlier parts of the text that Didymos could have been born flattened. To address this comment, we have re-affirmed this possibility in the bracketed text (line 328-329): “(to explain Didymos' squashed appearance if it were not born so)”.

Line 320: It is a speculation that is not addressed in the previous text.

Response: In response to your comment, and one made by reviewer 2, we have now updated the sentence pointing out the observation of the triangle limb-shape of the asteroid discussed in Fig. 4 and linking it to Fig. 9b to make it clear what we are discussing here (line 331-332): *The triangular shape of the Didymos ridge (Fig. 2a) may have been created by the mass shedding event (Fig. 9b).*

Line 324: it is not clear to me what "are likely >15-m-diameter " means.

Response: Thanks for noticing this. With your above comment, we clarified this section by adding the following text (line 335-345):

Numerical simulations [40] show that mass shedding events are relatively efficient at forming asteroid satellites, requiring only ~ 2–3% of the Didymos volume to be shed to form Dimorphos. Such a volume of material would be equivalent to a 3–5 m deep global layer (this layer would be deeper if mass was shed from a localized area). This is insufficiently thick to wipe away the largest craters on Didymos with depth near ~15 m that define the age of Didymos. As a consequence, we would expect, if the bulk shape is maintained, that the largest craters on Didymos could be relatively well preserved following a mass shedding event. While we cannot rule out effects of image resolution, the difficulty in clearly identifying craters <100 m-diameter on Didymos may be the result of surface modification caused by the deformation, landsliding, and mass-shedding.

I found some typos in lines: 19, 50 (Fig. 2a?), 54 (missing comma?), 61 (missing space between value and unit, multiple times in the text), 155 (input), 267.

Inconsistency between near-Earth and Near-Earth.

Weird numbering of sections (Acknowledgements and further).

Response: We address all three of the above minor editorial issues. Thanks for finding them.

Reviewer #2 (Remarks to the Author):

This paper describes the first detail results of the geology on Didymos and Dimorphos studied by the images taken from DART and LICIACube spacecraft. The authors studied the shape and surface geology of both asteroids and determined the crater size frequency distributions and the boulder surface distributions. They also conducted numerical studies for the factor of safety on

Didymos, and found that the surface cohesion could be less than 1 Pa to reproduce the boulder tracks found on Didymos. Furthermore, according to the previous studies for the shape of asteroid Bennu, they suggested that the interior cohesion of ~ 10 Pa was necessary for surface sliding, mass shedding, and some interior deformation. Similar to the discussion of Didymos, they also estimated the surface cohesion of Dimorphos and it was less than 0.9 Pa. Based on the surface cohesion of both bodies, the authors applied a crater scaling relationship for a crater chronology on both asteroids. Then, they estimated the surface age of both asteroids from the crater size frequency distribution. Finally, they found that the surface ages of Didymos and Dimorphos were very young compared to the Baupistina family, and they speculated that this means that the both asteroids could be the latest of multiple asteroid generations stemming from the original planet. This paper is quite worthwhile for the advance of the study of rubble-pile bodies and binary asteroids although most of the results may be confirmed or improved by the related mission of HERA, if it is succeeded. Moreover, this paper describes the geological features on both bodies before DART impact, so we can obtain quite important information by comparing pre- and postimpact geology on these bodies after HERA mission is completed. I think that this paper newly clarified other aspects of rubble bodies compared to Bennu, Ryugu, and Itokawa. Therefore, I strongly recommend the editor to publish this paper without delay. My major and minor concerns to this paper are described as follows. I appreciate if the authors answer to these comments.

One of the most important results of this paper is that the authors estimated the surface cohesion of the two rubble-pile bodies, and they are very small < 1 Pa. So, I think that Fig. 8 and Fig. 9 are important figures to determine them in this paper. Especially, the numerical calculation of Fig. 8 is very impressive for me to predict the boulder tracks. But there are some assumptions such as density, friction angle, upper layer thickness and rotation rate in this calculation. I am wondering how the FS is sensitive to these parameters, e.g. how much dose the FS change when the density changes from 3080 to 2520 kg/m³ or the friction angle changes from 20 to 40 degree?

Response: Increasing the density to 3080kg/m³ has a fairly minor consequence to the findings here. The surface slopes decrease and the cohesion needed to prevent surface failure changes from 1 to 0.5 Pa. A decrease in density to 2520kg/m³ would increase slope angles, and the cohesions needed from 1 to 2 Pa. Friction angles of 25 degrees would call for more cohesion (We don't think 20 degrees is reasonable for any natural occurring soil). If you look at Barnouin et al. 2022 in JGR Planets you can understand the effects of friction angle: steeper angles require less cohesion but these values of cohesion do not change dramatically with friction angle. Regolith thickness is probably the biggest factor. A 3m layer of regolith would require a cohesion of 0.3 Pa. To first order, our finding that the surface of this asteroid must have a cohesion of ~ 1 Pa does not change much with all the variables mentioned. We have added the following phrase in italic to the previous sentence at lines 169-173.

“Thinner layers and a lower bulk asteroid density would require less cohesion [35], while thicker layers and larger densities (within ± 280 kg/m³) would require more cohesion but always < 2 Pa. For our nominal assumptions (density of 2800 kg/m³ and 10m thick regolith layer), surface $C > 1$ Pa would shut down surface displacements completely.”

In addition to Fig.8, Fig.9 is also impressive for me. But I cannot follow how to derive several curves on Fig.9 (a). Ref.41 is cited in this part, but I don't understand why 10 Pa cohesion is necessary in this model.

Response: We derive the failure curves in Fig.9(a) using the method described in Zhang et al. (2022; the original Ref. 41). Each failure behavior is associated with a specific failure condition and is represented by an independent failure curve. For the surface landslides (the orange curve) and mass shedding (the blue curves), we use surface slope distributions and evaluate failure conditions for different near-surface cohesions. For the interior deformation (the red curve), we use the interior structural stress state distribution derived from a semi-analytical stress model to evaluate what cohesion failure (deformation) occurs. The structure of Didymos would undergo the corresponding failure behavior if its surface/interior cohesion is smaller than the value given by each curve. Therefore, at Didymos' current spin period of 2.26 hr., as indicated by the cross-over of the red curve and the green vertical line in Fig.9(a), maintaining ~10 Pa of interior bulk cohesion is necessary to prevent its internal structure from deformation. ~1Pa at the surface leads to landsliding.

We have updated the caption to Figure 9 (new text in *Italic*):

Failure and reaccumulation of Didymos. (a) Failure-mode diagram of Didymos for a density of 2800 kg/m³. These curves mark the minimum cohesion required to prevent structural failure in Didymos across different spin periods. The analysis considers three potential failure modes common to rubble-pile objects. Each curve is derived based on the corresponding failure criteria. Interior deformation that affects the entire asteroid occurs if C is below the red curve, which for Didymos...

Minor comments:

1. Figure 3: There is no explanation for Fig. 3(a). Only (b) is described.

Response: This is now addressed in response to this comment and reviewer 3. See new Figure 3 caption reads:

Composite images of Dimorphos (DRACO image collected on Sept 26, at 23:14:09 UTC; a) and Didymos (LUKE images collected on Sept 26, at 23:17:27; b). The images are composites of two stretches to show the full outline of the bodies. Regions of Dimorphos illuminated by Didymos shine appear light-grey in (a). Arrows in (b) point to the dark limb of Didymos back-lit by DART ejecta. In (b), the +Z (N) direction is tilted by a few 10s of degrees into the page. Dimorphos and DART ejecta can be seen in (b) at roughly 8:30 clock position from the center of Didymos.

2. Line 42: “gsd” is not so common word. Please explain what “gsd” is.

Response: Thanks for catching this. We rephrased to refer to the pixel scale of images, rather than GSD.

3. Line 47-48: The following sentence is obscure, “The visual identification ... on both

asteroids”. Could you please rewrite clearly?

Response: Clarified now with (line 48-50) “*The terrains identified are consistent with the spatial distribution of surface roughness measured on Didymos [14].*”

4. Line 68-71 and Fig.5: These sentences indicate that the boulder number density decreases with the increase of the latitude. I am wondering what materials occupy the inter space among boulders at high latitude. Is the space among boulders occupied by grains smaller than 0.5m?

Response: We no longer think this is true. The total number of boulders are identical at all latitudes. See our response to reviewer 1 for more details. For boulders that a statistical approach shows we identify completely, we do not see a latitudinal or longitudinal dependence on their numbers (see updated Fig. 5).

5. Fig.6: I cannot imagine how to prepare Fig.6(c). What is the definition of slope? Is it an average of some area?

Response: We compute the slope of each facet by using the dot product of the normal vector and the local gravity vector. We weight the slope of each facet by their surface area in order to show a slope distribution representative of what any region on the asteroid has. The peak in the distribution shows the average area weighted slope of the oblate ellipsoid comprising Dimorphos. The tail beyond 10 degrees is primarily due to the steep slopes of the boulders that are captured in the shape.

We have written some clarifications in the Fig. 6 caption: *(c) The global distribution of slopes weighted by surface area.*

6. Line 89: Could you please explain “local debris aprons”? I don’t see any apron in the photo.

Response: This is an accumulation of rocks leaning or laying against a large boulder. They are pointed out by the arrows. We clarified the text now (line 87-90):

Our preferred interpretation, therefore, is that many of these boulders leaning or laying on other boulders are a type of local debris aprons, where rocks have just slid off larger boulders and accumulated by their side with no preferred orientation. These rock accumulations are not an indication of widespread regional regolith transport.

7. Figure 7(a): What is μ in the caption? There is no definition of μ .

Response: μ is now removed. We define dashed red line (a) in a new caption for Figure 7:

Geometric crater depth to diameter ratio (a ; see [26] for definition) measured on Dimorphos, and crater size frequency distribution on Didymos (b) and Dimorphos (c). Dotted red-line shows (a) d/D results for Dimorphos from DTMs using the latest GDTM [10]. Didymos’ GDTM was of insufficient quality to measure reliable d/D .

8. Line 155: “nputs” may be typo. It may be “inputs”.

Response: Fixed

9. Figure 8: There is no number “a” in the top figure of Fig.8. The caption “C=0.5 Pa” of Fig.8f may be “C=1 Pa”.

Response: Both these issues are now fixed. Thanks for noticing.

10. Line 177-178: What are the mass concentrations which induce the protrude? I cannot image it. Is it large blocks without porosity?

Response: Yes. That is basically what we think these are. We have added this to the main text to clarify (line 180): *large distinct mass concentrations (large blocks)*

11. Line 184-185: Theses sentences may have a contradiction because it says that the strength is needed for mass-movements and mass shedding but the interior deformation is needed for the flattening. Could you please explain it more easily?

Response: Spinning asteroids require some strength to allow mass shedding – if they didn’t have that strength, the asteroid would deform into a pancake and then eventually disrupt. However, this strength can still be small enough to allow some deformation.

We clarified the paragraph where this line is located to address this confusion (line 188-206):

The presence of 10 Pa strength within Didymos is consistent with the evidence for surface mass-movements and mass shedding. Several studies show that asteroid spin-up facilitates mass movement and mass shedding, especially for lower porosity objects like Didymos, if the interior possesses some strength [e.g., 43, 46]. The interior strength of Didymos, however, must not be so large that it prevents the asteroid from flattening (unless the asteroid were formed in a flattened state to begin with). To permit such internal deformation while also shedding, asteroid spin-up calculations first used at Bennu [43], but adapted for Didymos show that an interior average bulk C of ~10 Pa (shear stress level at zero pressure) is required near the current spin-rate for a bulk density of 2800 kg/m³ and a friction angle of 35° (Fig. 9). The deformation would permit the observed flattening of Didymos and the non-circular equatorial circumference, as any interior mass concentration shift outwards. Anything stronger than the C of >20 Pa would prevent any interior deformation until Didymos spins up to <2.21 hours. Interior strengths similar to the surface C ≤ 1 Pa would see the entire asteroid deform into a near pancake (which is not observed), before mass shedding begins. It should be noted that at the center of Didymos, the overburden stresses would exceed 100 Pa, and friction effects, rather than cohesion, dominate. The spin-up calculations also indicate that 10 m-radius boulders would need to travel more than 100 m along the surface to possess enough momentum to be ejected and enter orbit.

Line 196-198: I cannot understand the following sentences, “The deformation calculations ... and enter orbit”. Please make clear. What is the deformation calculation and so on?

Response: To address this comment, we have clarified what we meant by “deformation calculation by using instead “spin-up” calculation in this line (now line 204). We define what we mean by ‘spin-up’ calculation further up in the text (line 194): *To permit such internal deformation while also shedding, asteroid spin-up calculations first used at Bennu [43], but adapted for Didymos show that an interior average bulk C of ~10 Pa (shear stress level at zero pressure) is required near the current spin-rate for a bulk density of 2800 kg/m³ and a friction angle of 35° (Fig. 9).*

Figure 9(b): This figure is so impressive to explain the assumption that produce the triangular ridge and mass shedding. I recommend the authors to discuss this figure in the main text.

Response: In response to this and reviewer 1 comment, we have now added a reference to Fig 9b in the main text and added a reference to Fig 4a where we show the triangular shape limb of Didymos. See line 331-332 *The triangular shape of the Didymos ridge (Fig. 2a) may have been created by the mass shedding event (Fig. 9b).*

Line 234: “low C (le 0.9 Pa)” may be typo. It may be “low C (<0.9 Pa)”.

Response: We fixed the text. It was indeed a typo (now line 241).

15. Line 274: What is the origin of the width of a surface age, 0.09-0.3Myr? Do you use the different strength to obtain 0.09 and 0.3 Myr?

Response: We have clarified what we did and referenced Figure 7c: We used the lower range of surface strength inferred from our analysis and initial beta estimates because the craters we have do not penetrate deeply in Dimorphos, and more likely encounter a weaker surface. See line 278-285:

In the case of Dimorphos, the craters are small (≤ 10 m) in diameter suggesting that their impact energy is not vcery high. The impactors for these craters probably never penetrated below a few meters in depth, and so are less likely to have encountered a stronger substrate inferred by the lineaments. . The lower-end of strength (0–0.1 kPa) estimates determined by numerical simulations of the DART impact [32–34] and inferences from Ryugu are, therefore, preferred. Consequently, the surface age of Dimorphos must also be well-modeled by impacts in a weak (near the gravity regime) target, with a surface age of 0.09–0.3 Myr (Fig. 7c)

Line 280: Why does the author adopt 5m for the limit of armoring effect?

Response: We have now clarified the text now. See line 289-296: *As the number density of boulders on Dimorphos is larger than on Bennu [11], we might expect that the transition into the armoring regime occurs at a larger diameter; maybe even at ~5 m where we see the break in slope of the Dimorphos CSFD (Fig. 7). But this transition could be offset by the greater yield strength of individual S-type boulders relative to more crushable C-type ones that may reduce the efficiency of armoring [58]. The exact consequence on the true surface age of Dimorphos is*

therefore not exactly known and is likely a second-order effect compared to the poorly understood cratering strength.

Line 394-395: The following sentence is obscure, so please rewrite it clearly, “Intermediate to small craters ... below others”. What is “others”?

Response: This was revised (line 406-407): *Intermediate to small craters on Didymos should be present below some steep slopes, but heavily modified below others.*

Line 417: “global datasact” may be typo. It may be “global dataset”

Response: We fixed this typo (line 429).

Reviewer #3 (Remarks to the Author):

This is an interesting paper detailing primary characteristics of the Didymos-Dimorphos binary system prior to the DART impact. Results here will be of broad relevance to NEO and asteroid studies, in particular, estimates of material strength and surface/formation ages here provide key constraints on system origin mechanisms and the overall (extremely young) system age. Methods and approaches utilized are generally state-of-the-art and well explained. I recommend its publication after addressing the minor issues indicated below.

lines 17-18 For the non-specialists, please clarify the definition of intermediate, major, minor, and z axes. It is notable that both objects have equatorial circumferences that are much closer to circular than most small bodies, i.e., they are much less ellipsoidal and are closer to oblate spheroids with a b/a near unity. This seems inconsistent with the tone of the first sentence in the abstract that instead emphasizes the equator of Didymos as being non-circular, when it instead appears to be remarkably near-circular.

Response: We see what look like N-S longitudinal ridges in some of the images of Didymos. We have modified the abstract to clarify what we mean, both in response to this reviewer’s comment and reviewer 1. The abstract now reads: *“These images reveal that the primary asteroid, Didymos, is flattened and has plausible undulations along its equatorial perimeter.”* This address both reviewer 1 and 2 concerns. We have also added a schematic in Figure 1 to make clear what we mean by the three axes, a, b and c, and clarify in the main text that these are the axis to the best fit ellipsoids: *“Using best fit ellipsoids, Didymos has an intermediate to major axis ratio $b/a = 0.96$, and a minor to major axis ratio $c/a = 0.73$.”*

line 17 Text is missing in parentheses after Fig. 1 reference

Response: This is now fixed (see line 17).

Fig. 1 caption: Are the terms “flattened” and “oblate” meant to convey different things? If yes, please clarify, and if not, use one term rather than both. Dimorphos seems similarly flattened, i.e., it has a similar c/a , to the secondaries in the plot, which seems to contradict the statement in

the caption. What is notable is its high b/a , i.e., its more circular equator compared to other secondaries.

Response: We have removed the use of “flattened” and “oblate”, and just highlighted the differences in the axis ratio of Didymos and Dimorphos to other primaries and secondaries.

Fig. 3 caption: Please insert a size scale bar. What is the white region on the right side of (a)?

Response: We have added a white bar and clarified what the light-grey/white region is in Fig 1a. See updated figure caption.

Line 61-63: 100-6 is an odd description; does this mean 6 to 100 cm in length? Please justify why is a 16 m boulder is “too big to have been formed by local impacts”.

Response: We have clarified this section by adding the following sentence (line 63-65): *The largest of the observed boulders is 16 m in length, too big to have been formed by the craters observed on Dimorphos, which are either similar or smaller than this length.*

Line 155: nputs \diamond input

Response: Now line 153. We edited the line as recommended.

Line 161: This is a surprisingly high density for a rubble pile, please comment.

Response: We comment on this in the discussion section (see lines 310-323). This is the current best estimate for the density for Didymos, with the caveat that the shape of Didymos may be too small, and there are ongoing efforts to refine the mass of Didymos. Such a density (low porosity) could be justified if the asteroid has large interior aggregates, and or fines that accumulate between voids at depth.

Line 175 “non-circular perimeter at Didymos”: Didymos is quite nearly circular at its equator ($b/a = 0.96$). Can the needed strength to account for $b/a = 0.96$ be quantified, and is this consistent with the estimates based on surface sliding and mass shedding cited on line 190?

Response: We have now clarified that protrusions in Didymos’ perimeter, that may be N-S ridges are likely indicators of some form of mass concentration within the asteroid. These mass concentrations are typically stronger (see lines 177-182).

Line 205: an near \diamond a near

Response: Replaced.

Line 293-294: Is it possible to use the inferred age differences to place any meaningful limit on the radiation torques that led to the spin up of the primary?

Response: Given the hypothesis given at the end of the manuscript, we do not think we can provide constraints on the radiation torques until Hera arrives. Further, there are just too many degrees of freedom since we also have to think about tides and BYORP which can add/subtract angular momentum. But we appreciate the thought.

Line 364-365: The accretion time of boulders in orbit around Didymos onto Dimorphos would seem to be extremely short. Is the tidal locking timescale short enough to explain why boulders are enhanced at low latitudes on Dimorphos, as suggested here? Or is it necessary to assume multiple mass shedding events from Didymos, with tidal locking occurring on much longer timescales between such events, as is described in hypothesis # 2 in lines 389-395?

Response: Since submitting this manuscript, we determined that only boulders $\geq 2\text{m}$ were completely observed, and that our inclusion of boulders $< 2\text{m}$ could have been biased by observation conditions. Using this new constraint, we find no dependency of boulder totals on longitude and latitude, which actually works much better with numerical models done by co-author Agrusa, which show that non-principal axis rotation is more likely than not when the secondary first forms. This implies that multiple mass shedding events may not be necessary, although cannot be eliminated because it remains difficult to generate Dimorphos' oblate shape.

These implications are now located in lines 363-376.

Further, the lack of any preference in the total number or size of boulders with latitude and longitude, the presence of an intimate mixture of boulders and cobbles, and the lack of obvious fine regolith are all consistent with Dimorphos being the result of mass shedding from Didymos and gravitational re-accumulation. Material shed from Didymos is expected to form a ring of material that includes all sizes around Didymos [43, 45, 69], except for the very finest ($< 100 \mu\text{m}$) particles. These finest particles get quickly swept away by solar radiation forces [e.g., 70], within hours as exemplified by observation of the DART tail [54]. The rest of the rocks accumulate quickly, in a handful of days. Their random distribution, regardless of latitude or longitude (Fig. 5) is consistent with models of secondary formation using large discrete element particles [40] where the satellite often forms with a non-principal axis rotation, and can accrete material at all latitudes and longitudes. These large particle models, however, also usually generate an elongated secondary, which Dimorphos was not. Some additional accumulation process, therefore, may be required to explain its oblate shape.

REVIEWERS' COMMENTS

Reviewer #1 (Remarks to the Author):

I would like to express my gratitude to the authors for their thorough revision of the manuscript based on the feedback provided by the referees. Addressing the comments and incorporating suggestions has significantly strengthened the quality and clarity of the manuscript.

I do not have additional comments.

Reviewer #2 (Remarks to the Author):

I appreciate the authors to improve their manuscript according to my comments, and I am satisfied with their answers to my comments and their revision. Therefore, I recommend the editor to publish this paper.

I don't have any comments and questions on this revised manuscript.

Reviewer #3 (Remarks to the Author):

The authors have responded to all of my concerns and I support publication of the current manuscript.